# Unlocking The Power Of Layer-By-Layer Training And Fine-Tuning

**Liron Gelbard**                                                      *lgelbard@qti.qualcomm.com*
*Qualcomm Technologies, Inc.*

**Shay Landis**[*]                                                      *slandis@qti.qualcomm.com*
*Qualcomm Technologies, Inc.*

**Assaf Touboul**[†]                                                    *atouboul@gmail.com*
*Qualcomm Technologies, Inc.*

**David Yannai**                                                        *dyunusov@qti.qualcomm.com*
*Qualcomm Technologies, Inc.*

**Reviewed on OpenReview:** *https://openreview.net/forum?id=p5ObETPuTi*

## Abstract

Layer-wise (LW) and segmented training reduce memory by restricting gradient propagation, but often suffer convergence degradation. We propose *Segmented Propagation (SegProp)*, which keeps a small, trainable *global head* (final layers + task head) active on the loss path throughout training, while updating only the current segment plus this shared head at each stage. This induces depth-wise gradient sparsity and reduces peak activation/optimizer footprint. Empirically, SegProp substantially closes the LW vs. End-to-End (E2E) gap on ResNet-18/50 for CIFAR-10 and achieves competitive performance under harder ImageNet-scale training with ViT, quantifying a clear accuracy–time–memory frontier as global-head depth and segmentation granularity vary. We further provide a system-level feasibility study on LLaMA-70B with $8 \times 40$ GiB GPUs, showing that SegProp enables larger feasible batches than FSDP with CPU offload and characterizing the resulting compute–memory trade-off via a detailed FLOPs analysis. Finally, we show that, in the evaluated 7–12B fine-tuning setups, SegProp matches or nearly matches end-to-end fine-tuning across downstream evaluations.

## 1 Introduction

Training modern large language models (LLMs) and deep neural networks is increasingly limited by memory and communication. For billion-parameter LLMs, end-to-end (E2E) backpropagation often exceeds single-GPU memory even with techniques such as activation checkpointing and quantization (HuggingFace), pushing practitioners toward pipeline parallelism (Wang et al., 2025).

Layer-wise (LW) training, where blocks are optimized using local losses, reduces peak activation memory and offers flexible parallel scheduling (Bengio et al., 2006; Hinton et al., 2006). Yet it consistently underperforms E2E optimization in deep models (Sakamoto & Sato, 2024). Prior work links this gap to information loss: local objectives encourage intermediate layers to discard input information that remains useful for the final

---

[*]Corresponding author.
[†]Work conducted while employed at Qualcomm.

prediction (Sakamoto & Sato, 2024; Wang et al., 2021). Prior Hilbert–Schmidt Independence Criterion (HSIC) based analyses suggest that, under purely local objectives, intermediate representations may become less aligned with the end task as depth increases (Tishby et al., 2000; Tishby & Zaslavsky, 2015). A central issue is the lack of persistent global supervision: each segment is trained to satisfy its own auxiliary head rather than the model's final task.

We argue that segmentation itself is not the core problem; the missing piece is a persistent global target. We introduce Segmented Propagation (SegProp), a training paradigm that keeps a small *global head* (the model's final layers and task-specific head) active and trainable throughout training. Earlier segments are optimized one at a time, but always under a single task-level loss computed by this shared head. Lightweight training-only adapters bridge intermediate features to the head when needed and are removed at inference. This provides consistent task-level supervision across stages while retaining much of the memory and parallelism benefit of segmented training.

Our goal is to reduce peak training-time memory and optimizer state—enabling larger micro-batches or deeper models on the same hardware—without sacrificing E2E-level performance. Concretely, we:

- Propose Segmented Propagation (SegProp), which combines the efficiency of segmented / layer-wise (LW) training with persistent global supervision by maintaining a trainable global head and a single task-level loss across all segments.

- Introduce a practical two-stage procedure: (i) train a base prefix jointly with the global head; (ii) iteratively train intermediate segments while keeping the global head active, using lightweight adapters that are discarded at inference.

- Show that SegProp substantially narrows the gap between LW and end-to-end (E2E) training on CNNs and fine-tuning on LLMs: on CIFAR-10, SegProp improves ResNet-50 accuracy from 90.0% (LW) to 94.3%, approaching E2E at 95.5%; for Mistral-Nemo-Instruct-2407 and Llama-3.1-8B-Instruct, SegProp segmented fine-tuning matches or nearly matches E2E performance on MMLU and WinoGrande.

- Extend the evaluation to harder regimes by quantifying the accuracy–time–memory trade-off under ImageNet-scale ViT training as segmentation granularity and global-head depth vary.

- Provide a system-level feasibility and compute–memory analysis for LLaMA-70B on $8 \times 40$ GiB GPUs, including a detailed FLOPs characterization of the trade-offs relative to sharded E2E training with cpu offload.

Figure 1 contrasts standard LW training, E2E backpropagation, and SegProp.

## 2 Related Work

### 2.1 Layer-wise Training and Information Bottleneck

Layer-wise (LW) training was originally proposed to address the credit assignment problem and to provide improved initialization for deep networks (Bengio et al., 2006; Hinton et al., 2006). More recently, empirical studies indicate that LW training can suffer from performance degradation as depth increases, often quantified using the Hilbert–Schmidt Independence Criterion (HSIC) as a proxy for mutual information. Such degradation is associated with poorer generalization and diminishing accuracy gains in deeper models (Sakamoto & Sato, 2024; Wang et al., 2021). The information-bottleneck viewpoint has been used as an interpretive lens for such phenomena; we adopt it only as qualitative motivation rather than as a formal or measured quantity in our study.

A frequently cited limitation of standard LW training is the absence of persistent task-level supervision. In prior analyses, this is associated with representations that can become less useful for the final task as depth increases under purely local objectives.

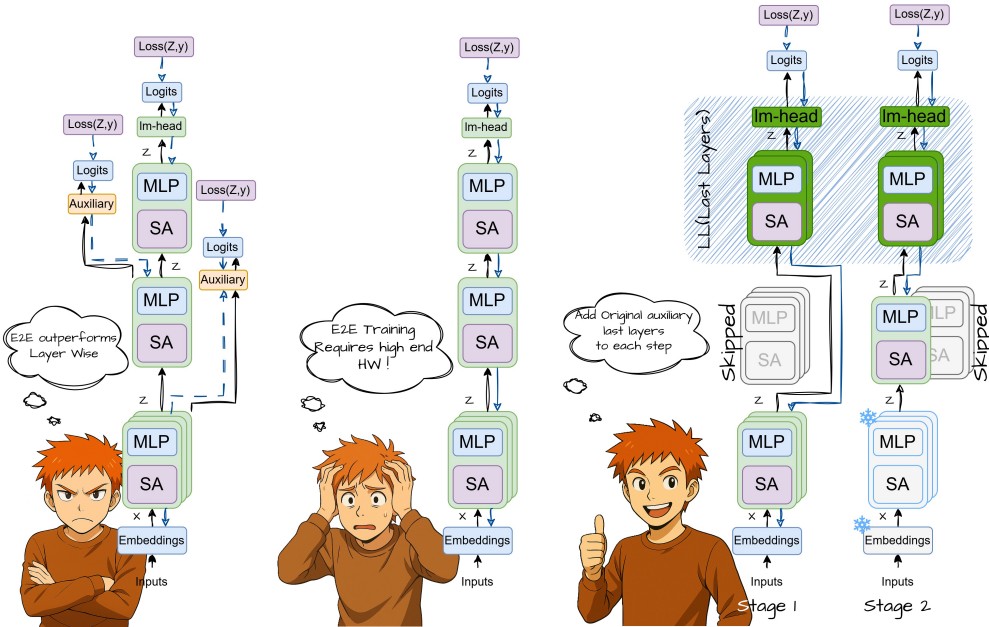

Figure 1: From left to right: layer-wise (LW), End to End (E2E), Segmented Propagation (SegProp).

We address these limitations by rethinking the role of the final layers in segmented optimization (see Figure 1). Specifically, we propose *Segmented Propagation (SegProp)*, which reintroduces the final layers during LW training as a shared global head. This design restores task-level information flow to earlier segments while retaining the efficiency benefits of segmented optimization. We hypothesize that persistent task-level supervision can reduce premature stage-local specialization; empirically, we observe improved intermediate-layer task performance across depth/stages.

We emphasize that we do not provide a formal information-theoretic analysis of this effect; rather, we demonstrate an empirical training design that consistently reduces the LW–E2E gap across the evaluated settings.

## 2.2 Biologically Plausible and Modular Training

Alternatives to backpropagation, such as Hebbian learning (Hebb, 1949), reservoir computing (Bianchi et al., 2020), and signal propagation (Kohan et al., 2022), aim to reduce memory usage and computational cost by localizing learning. Modular and block-wise training strategies (Belilovsky et al., 2018; Gomez et al., 2022) seek to balance parallelism and accuracy, but they often rely on backpropagation within blocks. The Forward-Forward algorithm (Hinton, 2022) enforces distinct roles for each layer, but it still suffers from information loss as depth increases.

## 2.3 Activation Checkpointing and Memory-Efficient Training

Checkpointing reduces peak memory by storing a subset of activations and recomputing the rest during the backward pass, trading compute for memory (Chen et al., 2016; He & Yu, 2023; Purandare et al., 2023; Korthikanti et al., 2022). It is a key tool for scaling deep models under constrained resources (Sakamoto & Sato, 2024). Activation checkpointing (AC) and selective AC (SAC) are standard: SAC applies checkpointing only to critical layers to balance recomputation overhead against savings. While effective, recomputation can add latency and bandwidth pressure, especially in distributed or multi-GPU settings.

**SegProp memory model.** In SegProp, only the active segment and the shared global head require gradients and optimizer state; all other segments run forward-only, reducing peak activation memory and optimizer footprint while preserving a single task-level loss via the persistent head.

**Snapshot checkpointing (SnapCheck).** Complementary to AC/SAC, SnapCheck caches detached activations from the frozen prefix and reuses them across iterations, reducing redundant forward compute without increasing peak memory. It integrates naturally with SegProp and is used only during training.

## 2.4 Progressive Growth Transformers (PGT)

A recent line of work proposes constructive scaling of Transformer language models by progressively growing depth on top of a frozen substrate, rather than training a fixed deep architecture monolithically. In particular, Bochkov (2025) introduce a progressive layer-wise growth procedure in which a shallow model is trained to convergence, then its trained layers are frozen, a new Transformer block is stacked on top, and training continues by updating only the newly added layer; this process is repeated to incrementally increase depth. This methodology is motivated by the hypothesis that high-level semantics are an emergent property of deep Transformer composition and thus can be constructed incrementally on a stable, frozen foundation. "

**Comparison to Segmented Propagation (SegProp).** Both PGT and SegProp employ stage-wise training with freezing, but they enforce different supervision structures. SegProp is a segmented optimization paradigm designed to maintain persistent task-level supervision: it keeps a shared global head (final layers + task head) on the loss path throughout training. In contrast, PGT's primary lever is architectural growth under a fixed trainable-parameter budget: during growth stages, only the newest layer is trainable while the existing stack remains frozen, and the authors optionally interleave global readjustment via LoRA-based tuning phases under a constant training budget.

**Head training and persistent supervision.** An important practical distinction concerns the role of the output head during staged training. SegProp explicitly keeps the global head trainable at every stage to ensure consistent task-level supervision across segments. For PGT the output head is not jointly trained as a persistent global head during growth stages, with global coherence instead supported by separate LoRA-based tuning phases.

## 2.5 Layer-selective fine-tuning

Recent work improves memory-efficient LLM adaptation by updating only a subset of layers. LISA uses importance-sampled layer freezing during AdamW-style updates and reports strong instruction-tuning results at low memory cost (Pan et al., 2024). OWS similarly samples layers, prioritizing layers with more outliers and optionally incorporating low-rank gradient projection, demonstrating gains across LLaMA/Mistral backbones (Li et al., 2025). These methods are complementary to SegProp: they focus on which layers to update, whereas SegProp focuses on how to stage optimization while preserving a single global end-task loss via a persistent head. The approaches are also potentially composable, e.g., applying LISA/OWS-style layer selection within the set of trainable parameters in SegProp (the active segment and/or the persistent head).

## 2.6 Memory-efficient full-parameter optimization

BAdam (Luo et al., 2024) proposes a memory-efficient approach to full-parameter LLM fine-tuning by combining block coordinate descent (BCD) with Adam-style updates, updating one parameter block (often a layer) at a time. This block-wise update schedule reduces the simultaneous optimization footprint compared to standard full-model Adam-style fine-tuning (Luo et al., 2024). While BAdam and our method both employ block-restricted updates, they differ in mechanism and goal: BAdam primarily changes the optimizer/update strategy via BCD-style block updates, whereas SegProp is a segmented training paradigm that keeps a persistent global head on the end-task loss path and trains one segment jointly with this head at each stage to preserve global supervision across stages. These approaches are potentially composable, e.g., using a block-wise optimizer within the parameters trained by SegProp (the active segment and/or the persistent head).

# 3 Segmented Propagation (SegProp)

## 3.1 Problem Setting

As previously noted, end-to-end (E2E) training of large models demands significant GPU memory: it must retain weights, activations, optimizer states, gradients, and other intermediates across the entire depth, leading to high peak memory use and extended runtimes.

To mitigate this burden, layer-wise (LW) training updates one layer or sub-block at a time rather than performing full-depth backpropagation (Bengio et al., 2006; Hinton et al., 2006). By restricting gradient traversal to the *active* segment and running earlier/later segments forward-only, LW substantially reduces the peak activation footprint and compute per step. However, because supervision is local to the active segment (often via an auxiliary head), LW can suffer information loss and suboptimal performance: intermediate representations may be insufficiently informative for the final task (Sakamoto & Sato, 2024). In agreement with prior results on convolutional architectures (ResNet-18/50, VGG11) showing sizeable accuracy degradation under LW (Sakamoto & Sato, 2024), our experiments on a Transformer-based architecture fine-tuning reveal an even more severe performance drop under MMLU and HumanEval+ (HE+) (Liu et al., 2023).

E2E training remains the standard approach: the network runs forward to produce outputs, the loss is computed, and stochastic gradient descent (SGD) applies the chain rule to propagate gradients through *all* layers in reverse order.

LW training, in contrast, proceeds stage-wise. For the first segment, the model runs forward through that segment and a small auxiliary module that aligns its output to the task objective; the segment is optimized against a local loss while the rest of the network remains frozen. Subsequent stages repeat this pattern for each segment: the network runs up to the current segment, a geometry-matching auxiliary module produces the supervision signal, the loss is computed, and only the active segment is updated. Auxiliary modules are training-only and discarded at inference.

We propose Segmented Propagation (SegProp): a segmented optimization method that retains the memory and scheduling advantages of layer-wise (LW) training while maintaining persistent task-level supervision. Concretely, SegProp keeps a shared global head—the final few layers together with the task head—active and trainable throughout training, and updates only the current backbone segment jointly with this head at each stage. The active segment is optimized using a single end-task loss computed by the shared head, using lightweight training-only adapters when needed to match intermediate feature geometry, which are discarded at inference. By keeping the same end-task objective on the loss path across stages, SegProp consistently reduces the LW–E2E gap in the evaluated settings while retaining most of the peak-memory benefits of segmented training. Empirically, SegProp approaches end-to-end accuracy for CNN training from scratch (ResNet-18/50 on CIFAR-10), extends to ImageNet-scale ViT training where it exposes an explicit accuracy–time–memory trade-off as segmentation and global-head depth vary, and matches or nearly matches E2E quality in 7–12B LLM fine-tuning; we further provide a system-level feasibility and FLOPs analysis for LLaMA-70B on $8\times40$ GiB GPUs that characterizes the resulting compute–memory trade-off under sharded training.

## 3.2 Two-Stage SegProp Strategy

SegProp employs a two-stage training strategy designed to balance memory efficiency with strong supervision:

1. **Joint training of a base prefix with the last layers.** Select a prefix of the model (base layers) and train it jointly with a small set of final layers (the *last-layers* module) to establish a strong task-level signal early. Intermediate layers are skipped to reduce compute and memory. Gradients flow through the base prefix and last layers only; earlier/later layers are non-trainable in this stage.

2. **Iterative layer-wise training of intermediate layers with global supervision.** Train each intermediate layer individually alongside the last-layers module, which provides a consistent, task-level loss across stages. Previously trained layers remain frozen; only the current target layer and

the last layers receive updates. After a layer is trained, its weights are committed (frozen) for the subsequent iteration. Lightweight adapters (training-only) are used as needed to match geometry between the current layer's output and the last-layers input; these adapters are discarded at inference.

### 3.3 Formalization

We first describe SegProp in an architecture-agnostic way for a generic deep network, and then instantiate it for Transformers and ResNets in the experiments.

**Model and segmentation.** Let $\{(x_i, y_i)\}_{i=1}^m$ be a dataset with inputs $x_i \in \mathcal{X}$ and targets $y_i \in \mathcal{Y}$, and let $f_\theta : \mathcal{X} \to \mathcal{Y}$ be a deep network with parameters $\theta$. We view $f_\theta$ as a composition of $n$ trainable blocks (layers or segments):

$$f_\theta(x) = h \circ f_{n-1} \circ f_{n-2} \circ \cdots \circ f_0(x), \tag{3.1}$$

where $f_j$ denotes the $j$-th block (e.g., a ResNet stage or a Transformer decoder layer) and $h$ is the final task-specific head (e.g., classifier or LM head).

For compactness, define the half-open composition

$$F_{a:b}(x) := f_{b-1} \circ f_{b-2} \circ \cdots \circ f_a(x), \quad 0 \le a < b \le n, \tag{3.2}$$

with the convention $F_{a:a}(x) = x$.

We choose:

- a *base prefix* depth $p \in \{0, \ldots, n-1\}$,
- a *last-layers* size $r \in \{1, \ldots, n-p\}$.

The base prefix and last-layers module are then

$$f^{[0:p]}(x) := F_{0:p}(x), \tag{3.3}$$

$$\mathcal{LL}^{(r)}(z) := h \circ F_{n-r:n}(z), \tag{3.4}$$

where $F_{n-r:n}$ denotes the composition of the last $r$ blocks preceding the head $h$. Intuitively, $\mathcal{LL}^{(r)}$ is a small *global head* consisting of the head $h$ and the last $r$ blocks of the network. All remaining blocks, with indices $j \in \{p, \ldots, n-r-1\}$, are treated as *intermediate* and will be trained iteratively.

We denote by $\mathrm{StopGrad}(\cdot)$ an operator that prevents gradient propagation through its argument (e.g., `detach()` in PyTorch), ensuring that gradients do not flow into earlier segments when we train a given block.

**Stage 1: joint training of base prefix and global head.** In Stage 1, we train the base prefix $f^{[0:p]}$ jointly with the last-layers module $\mathcal{LL}^{(r)}$, while skipping intermediate blocks.

Given a minibatch $\mathcal{B} = \{(x^{(b)}, y^{(b)})\}_{b=1}^{|\mathcal{B}|}$, the forward pass is

$$z^{(b)} = f^{[0:p]}(x^{(b)}), \tag{3.5}$$

$$\tilde{z}^{(b)} = A_p(z^{(b)}), \quad \text{(optional adapter; identity if not needed)} \tag{3.6}$$

$$\hat{y}^{(b)} = \mathcal{LL}^{(r)}(\tilde{z}^{(b)}), \tag{3.7}$$

with task loss

$$\mathcal{L}_1 = \frac{1}{|\mathcal{B}|} \sum_{b=1}^{|\mathcal{B}|} \ell(\hat{y}^{(b)}, y^{(b)}), \tag{3.8}$$

where $\ell$ is the end-task objective (e.g., cross-entropy). Backpropagation updates only the parameters of $\{f^{[0:p]}, \mathcal{LL}^{(r)}, A_p\}$; all other blocks are frozen. After convergence, we *commit* (freeze) the base prefix $f^{[0:p]}$, while keeping $\mathcal{LL}^{(r)}$ active and trainable for Stage 2.

**Stage 2: iterative training of intermediate blocks with global supervision.** In Stage 2, we train the intermediate blocks one at a time under a *single* task-level loss provided by $\mathcal{LL}^{(r)}$. Let $\hat{p}$ index the current block in $\{p, \ldots, n-r-1\}$. For minibatch $\mathcal{B}$, the forward pass is

$$z_{\hat{p}}^{(b)} = f_{\hat{p}}\Big(\text{StopGrad}\big(F_{0:\hat{p}}(x^{(b)})\big)\Big), \tag{3.9}$$

$$\tilde{z}_{\hat{p}}^{(b)} = A_{\hat{p}}(z_{\hat{p}}^{(b)}), \quad \text{(adapter; identity if not needed)} \tag{3.10}$$

$$\hat{y}^{(b)} = \mathcal{LL}^{(r)}(\tilde{z}_{\hat{p}}^{(b)}), \tag{3.11}$$

and the loss is

$$\mathcal{L}_2 = \frac{1}{|\mathcal{B}|} \sum_{b=1}^{|\mathcal{B}|} \ell(\hat{y}^{(b)}, y^{(b)}). \tag{3.12}$$

Here, gradients are applied only to $\{f_{\hat{p}}, \mathcal{LL}^{(r)}, A_{\hat{p}}\}$; all blocks with indices $< \hat{p}$ have been committed and remain frozen, and blocks with indices $> \hat{p}$ are not part of the computation graph at this stage. After training block $\hat{p}$ to convergence, we commit $f_{\hat{p}}$, discard $A_{\hat{p}}$ (training-only), and move to the next block.

**Efficiency and global supervision.** At any time, SegProp activates gradients only through the base prefix (Stage 1) or a single intermediate block (Stage 2) plus the shared global head. This reduces peak activation memory and optimizer state relative to end-to-end backpropagation, where gradients must flow through all $n$ blocks. At the same time, the persistent last-layers module $\mathcal{LL}^{(r)}$ enforces a *single* task-level objective across all segments.

In our experiments, we instantiate this generic scheme with:

- Convolutional backbones (ResNet-18/50), where $f_j$ are backbone segments (ResNet stages) and $\mathcal{LL}^{(r)}$ reuses the final convolutional layers and classifier (Appendix B);

- Vision Transformers (ViT) on ImageNet, where $f_j$ are Transformer blocks and $\mathcal{LL}^{(r)}$ consists of the last few Transformer blocks and the classification head (Appendix F); and

- Transformer-based LLMs, where $f_j$ are decoder layers and $\mathcal{LL}^{(r)}$ consists of the last few decoder layers and the LM head (Appendix C), including a system-level study for LLaMA-70B (Appendix D).

### 3.4 Snapshot Checkpointing (SnapCheck)

To avoid redundant computation during Stage 2, we introduce *Snapshot Checkpointing* (SnapCheck), a cache of frozen-prefix activations that can be reused across layer iterations. After computing the committed prefix output (Eq. (3.13)), for a minibatch $\mathcal{B} = \{(x^{(b)}, y^{(b)})\}_{b=1}^{|\mathcal{B}|}$ we form

$$z^{(b)} = f^{[0:p]}(x^{(b)}), \quad \forall b \in \{1, \ldots, |\mathcal{B}|\}, \tag{3.13}$$

and store *detached* snapshots $\bar{z}^{(b)} := \texttt{StopGrad}(z^{(b)})$ in a memory-efficient buffer $\mathcal{S}$, indexed by the prefix depth $p$ and a minibatch identifier (e.g., dataloader index, seed). When fine-tuning a subsequent layer $f_{\hat{p}}$, instead of recomputing $f^{[0:p]}(x^{(b)})$ on every step, we retrieve the cached activation:

$$z^{(b)} \leftarrow \mathcal{S}[p, \texttt{batch\_id}], \tag{3.14}$$

falling back to on-the-fly computation and insertion if the snapshot is missing.

Because $f^{[0:p]}$ is committed (frozen) during Stage 2, these snapshots remain valid across iterations, eliminating repeated evaluation of the prefix. This targets reduction of per-step compute and wall-clock time, especially when the committed prefix is deep ($p \gg 0$). SnapCheck is complementary to activation checkpointing (AC/SAC): AC reduces the *activation memory* of the trainable suffix, while SnapCheck reduces *forward compute* by reusing frozen-prefix outputs. Snapshots are training-only; inference uses the standard forward $h \circ F_{0:n}$ without snapshots. In practice, SnapCheck stores detached prefix activations per batch

index and reuses them across iterations when only deeper layers are being updated, so the cost of the frozen prefix is paid once per batch per prefix depth.

A step-by-step pseudocode description of Segmented Propagation Stochastic Gradient Descent (SegProp-SGD), including the handling of adapters, snapshot checkpointing, and freezing policies, is provided in Algorithm 1 in Appendix A. In practice, this algorithm implements the two-stage procedure described above: (i) joint training of a base prefix with the last-layers module, followed by (ii) iterative training of intermediate layers under a persistent global head and a single task-level loss.

## 4 Results and Analysis

### 4.1 SegProp Training for ResNet-18 on CIFAR-10

**Overview.** ResNet-18 is trained with SegProp by splitting the backbone into segments optimized in stages while retaining a shared global head throughout training. The head is always in the loss path and updated at every stage; training-only adapters map segment outputs to the head's input geometry.

#### 4.1.1 Backbone Segmentation and Global Head

We use an ImageNet-style ResNet-18 with 224×224 inputs, partitioned into four segments (stem + layer1, layer2, layer3, and part of layer4). A small global head (final convolutional sub-layer plus 2-layer MLP classifier) remains shared across all stages, while lightweight training-only adapters map each segment's output into the head's input geometry (details in Appendix B).

In the notation of Section 3.3, ResNet-18 is decomposed into four segments $f_0, \ldots, f_3$ (stem + layer1, layer2, layer3, part of layer4), and the global head $\mathcal{LL}^{(r)}$ consists of the remaining layer4 convolution and the final MLP classifier.

#### 4.1.2 Layer-Wise Training Setting for ResNet-18

For LW training the backbone is split into four segments (seg1–seg4), each trained sequentially with its own auxiliary classifier, followed by a final 'fc' stage that trains only the global classifier with the backbone frozen. At stage k (1–4), all earlier segments are frozen and only the current segment and its auxiliary head are updated. Auxiliaries are lightweight Conv–BN–ReLU adapters that normalize intermediate features to a fixed 512×7×7 resolution, followed by global average pooling and a linear classifier. Full architectural details are given in Appendix B.

#### 4.1.3 Results

Figure 2 summarizes ResNet-18 training on CIFAR-10 using 400 epochs (more details in Appendix B). Panel (a) compares the final-stage trajectories of SegProp, LW, and E2E, showing best test accuracy of 95.23% (SegProp), 93.69% (LW), and 95.50% (E2E). Panels (b–c) concatenate the four sequential training stages into a single timeline: each stage is trained independently while earlier segments are frozen, so the curve should be read as a *piecewise* learning process rather than one continuous end-to-end run. The sharp changes in slope around stage boundaries reflect a deliberate change in which segment is trainable (and, for LW, a change in the auxiliary head used for supervision), rather than an instability of the optimizer. The dot in each stage indicates the best checkpoint within that stage.

### 4.2 SegProp Training For ResNet-50 On CIFAR-10

Figure 3 summarizes ResNet-50 on CIFAR-10. Panel (a) compares SegProp, LW, and E2E with best test accuracy of 94.34%, 90.04%, and 95.53%, respectively. Panel (b) ablates SegProp global head size in the final stage (1conv 91.47%, 2conv 94.07%, 3conv 94.34%). Panel (c) concatenates SegProp's stages (seg1 89.96%, seg2 92.47%, seg3 93.08%, seg4 94.34%). Dots mark best-epoch checkpoints. Here, 1 conv/2 conv/3 conv denote layers derived from the last convolutional layers of the original ResNet-50 and applied as part of global head (LL).

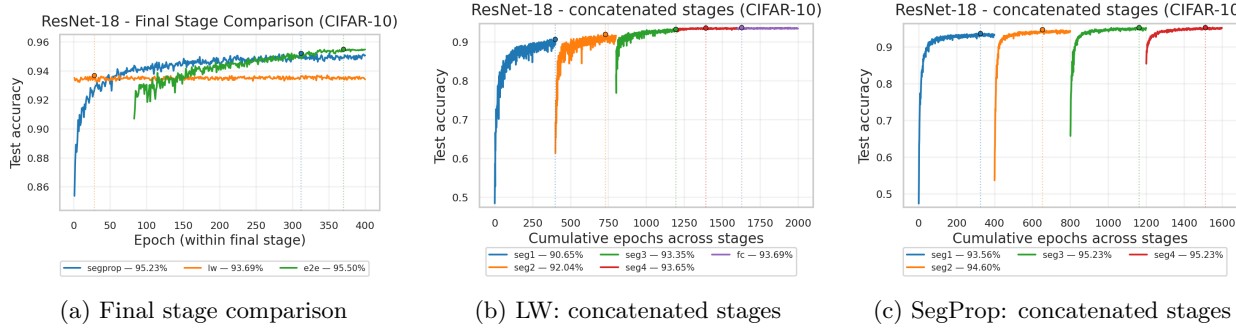

(a) Final stage comparison     (b) LW: concatenated stages     (c) SegProp: concatenated stages

Figure 2: ResNet-18 on CIFAR-10. (a) Final-stage trajectories comparing SegProp (best 95.23%), LW (best 93.69%), and E2E (best 95.50%). (b) LW training across stages. (c) SegProp training across stages. Dots denote best-epoch checkpoints.

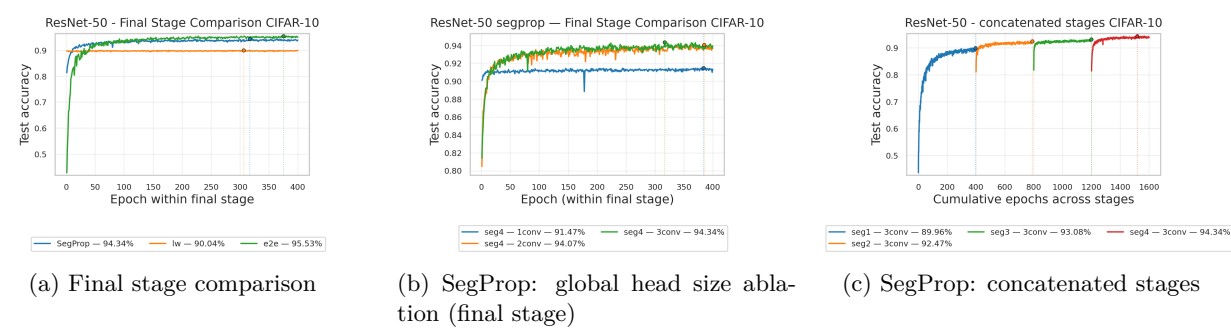

(a) Final stage comparison     (b) SegProp: global head size ablation (final stage)     (c) SegProp: concatenated stages

Figure 3: ResNet-50 on CIFAR-10. (a) Final-stage trajectories comparing SegProp (best 94.34%), LW (best 90.04%), and E2E (best 95.53%). (b) SegProp global head size ablation in the final stage: 1 conv 91.47%, 2 conv 94.07%, 3 conv 94.34%. (c) SegProp training across stages: seg1 89.96%, seg2 92.47%, seg3 93.08%, seg4 94.34%. Dots denote best-epoch checkpoints.

Figure 3 panel (c) visualizes SegProp as a staged optimization process in which each newly trained segment yields an incremental gain under the same persistent global head. The step-wise increases in best accuracy from seg1 to seg4 (89.96% → 92.47% → 93.08% → 94.34%) indicate that early stages learn broadly useful low/mid-level features quickly, while later stages provide smaller but consistent refinements as spatial resolution decreases and the representation becomes more task-specific. Importantly, because the global head remains trainable throughout, the gains compound across stages rather than being "reset" by changing supervision, which helps explain the large gap between SegProp and LW on ResNet-50.

### 4.3 Scaling SegProp to a Harder Vision Benchmark: ViT on ImageNet

The previous subsections showed that a persistent global head restores convergence and narrows the gap to end-to-end (E2E) optimization. We now extend this analysis to a more demanding vision regime: training ViT on ImageNet, where optimization is less forgiving.

**Setup.** We train ViT (details in Appendix F) under segmented optimization and report top-1 validation accuracy evaluated per epoch.

**Final-segment alignment (epochs 0–99).** Each segment is trained for a fixed budget of 100 epochs. To compare late-stage convergence across runs, we restrict each run to the final segment and re-index its last-segment epochs to a shared x-axis in $\{0, \ldots, 99\}$.

**Ablations: global-head depth vs. segmentation granularity.** Figure 4 summarizes two controlled comparisons in the final segment: (i) fixing the number of segments and varying global-head (GH) depth,

and (ii) fixing GH depth and varying the number of segments. To highlight small but consistent differences typical of ImageNet-scale top-1 accuracy, we zoom the y-axis around the peak accuracy within each panel.

**Accuracy–time trade-off.** Beyond accuracy, ImageNet training exposes non-trivial runtime differences across configurations. Table 2 reports the final validation accuracy at the end of the last segment together with training time. We observe that increasing segmentation yields substantial memory savings, while sufficiently deep global heads (e.g., GH=5) can recover vanilla accuracy at the cost of longer training time. Together, Figure 4 and Table 2 quantify the accuracy–efficiency frontier in this harder regime.

### 4.4   ImageNet-scale CNN memory and checkpointing

Table 1 reports peak GPU memory for ResNet-101 training on ImageNet-1k at a fixed global batch size of 1024 (8 GPUs × 128 per GPU). We split the *backbone* into `seg` sequential *segments* (trained stage-wise), while keeping a persistent global head (GH) of fixed depth 3 active throughout. In the table, `seg` counts only the backbone segments (excluding the GH). We report per-GPU peak memory and the corresponding multi-GPU total (computed as 8× the per-GPU peak).

**Checkpointing captures most of the peak-memory savings in this CNN regime.** A first key trend in Table 1 is that enabling gradient checkpointing (GC) yields a large reduction in peak memory for ResNet-101/ImageNet. For the Vanilla BP (E2E) baseline, enabling GC reduces the total peak from 153.79 GB to 73.17 GB (a 52.4% reduction). For SegProp with `seg`=2 (GH=3), the reduction is from 116.80 GB to 60.83 GB (47.9%). This pattern is consistent with an activation/graph-memory-dominated regime: checkpointing reduces peak memory largely by trading activation storage for recomputation in the backward pass.

**Segmentation provides additional (but diminishing) savings beyond checkpointing.** A second trend is that increasing segmentation reduces peak memory even when GC is disabled. With GH depth fixed to 3, moving from the Vanilla BP (E2E) reference point (153.79 GB total) to SegProp with `seg`=4 (99.54 GB total) yields a 35.3% reduction. The marginal gains diminish as segments become smaller (153.79→116.80→104.48→99.54 GB for Vanilla BP → `seg`=2 → `seg`=3 → `seg`=4). Intuitively, segmentation reduces the number of gradient-bearing blocks that contribute to the peak backward-state footprint at any given stage, complementing checkpointing but yielding smaller incremental gains once segments become short.

**Why this differs from large LLM training.** The ResNet-101/ImageNet results above illustrate a regime where activation/graph memory is a primary driver of peak usage, so checkpointing alone accounts for a large fraction of achievable savings. In large LLM training, however, peak memory is often dominated by optimizer state and sharding/state-management effects rather than activation storage alone; accordingly, our LLaMA-70B study focuses on depth-wise gradient sparsity and optimizer/state staging under modern distributed training paradigms, where reducing the number of gradient-bearing layers directly impacts the dominant memory terms.

**Contextual comparison to AugLocal.** AugLocal (Ma et al., 2024) reports ResNet-101/ImageNet memory decreasing from 157.12 GB (BP) to 97.65 GB (AugLocal), corresponding to a 37.9% reduction, while maintaining accuracy close to BP (77.34% vs. 76.70% top-1). Using Vanilla BP (GC off; 153.79 GB total in Table 1) as the internal reference, SegProp achieves a 35.3% reduction at `seg`=4 (99.54 GB total), i.e., $1 - 99.54/153.79 \approx 35.3\%$. Thus, in this ImageNet-scale CNN regime, SegProp's segmentation-driven memory reduction is of similar magnitude to AugLocal's reported savings, while Table 1 also shows that checkpointing can yield even larger reductions in this activation-dominated setting.

Table 1: **ResNet-101/ImageNet peak memory (global batch 1024, 8 GPUs).** We report per-GPU peak memory and the corresponding multi-GPU total ($8\times$ per-GPU). "Vanilla BP" denotes the non-segmented end-to-end baseline; rows below report SegProp settings with `seg` backbone segments (excluding the global head), with GH depth fixed to 3 blocks. GC denotes whether Gradient Checkpoints were enabled.

| Setting | GC | Peak/ GPU (GB) | Total (GB) |
|---|---|---|---|
| Vanilla BP (E2E) | off | 19.22 | 153.79 |
| Vanilla BP (E2E) | on | 9.15 | 73.17 |
| SegProp (`seg`=2, GH=3) | off | 14.60 | 116.80 |
| SegProp (`seg`=2, GH=3) | on | 7.60 | 60.83 |
| SegProp (`seg`=3, GH=3) | off | 13.06 | 104.48 |
| SegProp (`seg`=4, GH=3) | off | 12.44 | 99.54 |

Table 2: **ViT/ImageNet: final accuracy, training time, and peak GPU memory (final segment).** Final validation accuracy (top-1) is reported at the end of the last training segment (epochs aligned to 0–99). Training time corresponds to the trainer-reported total training time. Peak GPU memory is the maximum allocated GPU memory (GiB) across epochs.

| Configuration | Final Acc@1 (%) | Train Time (h) | Max GPU (GiB) |
|---|---|---|---|
| Vanilla BP (E2E) | 78.93 | 5.584 | 9.52 |
| SegProp (seg=2, GH=3) | 78.30 | 7.987 | 6.65 |
| SegProp (seg=2, GH=4) | 78.85 | 8.408 | 6.71 |
| SegProp (seg=3, GH=3) | 77.30 | 10.150 | 5.28 |
| SegProp (seg=3, GH=4) | 78.64 | 10.976 | 5.99 |
| SegProp (seg=3, GH=5) | 78.92 | 11.712 | 6.65 |
| SegProp (seg=4, GH=3) | 77.56 | 12.699 | 5.21 |
| SegProp (seg=4, GH=4) | 78.00 | 13.505 | 5.28 |
| SegProp (seg=5, GH=3) | 76.91 | 15.299 | 4.56 |
| SegProp (seg=6, GH=3) | 77.03 | 17.999 | 4.55 |

## 4.5 Memory Analysis: Regular FSDP vs. SegProp for LLaMA-70B

### 4.5.1 Setup

We consider training LLaMA-70B Touvron et al. (2023) with $L = 80$ decoder layers, hidden dimension $d = 8192$, distributed across $R = 8$ GPUs each with $M_{\text{GPU}} = 40\,\text{GiB}$ HBM (39.38 GiB total accessible). Parameters are stored in `bfloat16`; total parameter storage $\Phi \approx 141.2\,\text{GiB}$. All measurements are from confirmed successful runs on $8\times$A100-40 GiB GPUs. Three configurations at maximum feasible batch size are reported. In all cases, $T$=2048 tokens per sequence and $\approx$128 steps per epoch. Memory traces are collected over the first 10–15 steps of each epoch. Here $B$ denotes the per-GPU batch size and $\ell_{\text{GH}}$ the number of decoder layers in the Global Head (in addition to the LM head, corresponding to $\mathcal{LL}^{(r)}$ in the notation of Section 3.3). SegProp trains one body segment per epoch for 3 epochs (one per body segment), constituting 1 effective pass through the training data; FSDP trains for 1 epoch.

- **SegProp** $B$=13, $\ell_{\text{GH}}$=2: 3 body segments + GH, `FULL_SHARD`, optimizer CPU offload, gradient checkpointing, $\approx$128 steps/epoch, 3 epochs (1 effective epoch).

- **SegProp** $B$=13, $\ell_{\text{GH}}$=4: same as above but $\ell_{\text{GH}} = 4$.

- **FSDP+offload** $B$=9: `FULL_SHARD`, full CPU offload, activation checkpointing, $\approx$128 steps/epoch, 1 epoch.

Implementation and memory-management details for the 70B study are provided in Appendix D.

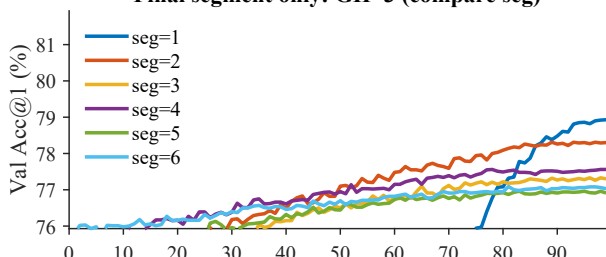

(a) Fixed number of segments, varying global-head (GH) depth.

(b) Fixed GH depth, varying number of segments.

Figure 4: **ViT/ImageNet: final-segment convergence under segmentation and global-head depth.** Validation Acc@1 is shown only for the last training segment, with epochs aligned to a common range of 0–99.

### 4.5.2 Regular FSDP

FSDP Zhao et al. (2023) with `FULL_SHARD` shards parameters, gradients, and optimizer states across all $R$ ranks. AdamW maintains two fp32 momentum buffers per parameter, totalling $2\Phi$ bytes. Per-rank memory at any point:

$$M_{\text{FSDP}} = \underbrace{\frac{\Phi}{R}}_{\text{params (bf16)}} + \underbrace{\frac{\Phi}{R}}_{\text{grads (bf16)}} + \underbrace{\frac{4\Phi}{R}}_{\text{opt. states (fp32)}} + \underbrace{2\frac{\Phi}{L}}_{\text{all-gathered layer}} + \epsilon \tag{4.1}$$

$$= 17.65 + 17.65 + 70.60 + 3.53 + \epsilon \approx 109.4 + \epsilon \text{ GiB}$$

The four terms are: sharded bf16 parameters ($\Phi/R$); sharded bf16 gradients ($\Phi/R$); sharded fp32 AdamW optimizer states—two momentum buffers at $2\times$ the bf16 size, giving $4\Phi/R$ (70.60 GiB); and the peak all-gathered parameters for up to two layers simultaneously during backward prefetch ($2\Phi/L$, 3.53 GiB). The optimizer states alone (70.60 GiB) exceed the 40 GiB GPU budget. **Regular FSDP cannot train LLaMA-70B on 8x 40 GiB GPUs without full CPU offload.**

**FSDP + full CPU offload.** With `fsdp_cpu_offload=True`, only the current layer's all-gathered parameters and activations occupy GPU memory.

$B=9$ ($T=2048$): peak alloc = 37.76 GiB, reserved = 38.19 GiB (0.11 GiB headroom). $B=10$ results in OOM; $B=9$ is therefore the maximum feasible batch size for FSDP.

### 4.5.3 SegProp

The number of gradient-bearing layers is:

$$\ell_{\text{grad}} = \ell + \ell_{\text{GH}} \approx \frac{L}{3} + \ell_{\text{GH}} \ll L \tag{4.2}$$

**Frozen-segment CPU offload.** After each frozen segment's forward pass, its sharded flat parameters are moved to CPU, saving 5.18 GiB per segment (measured). They are reloaded before the next step's frozen forward.

**Maximum batch size.** Attempting $B=14$ with $\ell_{\text{GH}}=4$ completes the forward pass (alloc = 20.038 GiB, reserved = 27.211 GiB after GH fwd, epoch 0) but runs out of memory during the backward pass of epoch 0 (seg0 training). The OOM is caused by *allocated* memory exceeding the 39.381 GiB GPU total during gradient computation. At the identical checkpoint (after GH fwd, epoch 0) for $B=13$: alloc = 19.069 GiB, reserved = 25.451 GiB; the backward pass completes successfully with peak alloc = 38.016 GiB and peak

reserved $= 38.268\,\text{GiB}$ ($1.11\,\text{GiB}$ headroom). Scaling from $B{=}13$ to $B{=}14$ increases activation memory by $\approx 7.7\%$, pushing the backward-pass peak allocated over the $39.381\,\text{GiB}$ GPU total. Hence $B{=}13$ is the maximum feasible batch size for SegProp with 3 segments and $\ell_{\text{GH}}{=}4$.

### 4.5.4 Measured Memory Trace

Tables 3 and 4 report measured memory at key checkpoints (first step of each epoch, rank 0), for FSDP $B{=}9$ and SegProp $B{=}13$, $\ell_{\text{GH}}{=}4$, respectively. The peak alloc/reserved column is the highest allocation/reservation reached at any point up to that checkpoint (cumulative maximum since the last reset).

Table 3: FSDP+offload, $T{=}2048$, 1 epoch. Peak stats reset at step start; all values are per-step peaks. $B{=}9$ **(15 steps):** peak alloc $= 37.76\,\text{GiB}$ (during BW), peak reserved $= 38.19\,\text{GiB}$ ($1.19\,\text{GiB}$ headroom) — succeeds. $B{=}10$ **(13 steps):** OOM during BW; last recorded peak alloc $= 36.67\,\text{GiB}$ (pre-BW), implying BW would require $\geq 39.38\,\text{GiB}$ — fails.

| Checkpoint | peak_alloc (GiB) | peak_reserved (GiB) | free (GiB) |
|---|---|---|---|
| *B=9, Epoch 0, step 0 (peak reset at step start)* | | | |
| Step start (peak reset) | 0.002 | 4.164 | 35.217 |
| After FW | 31.084 | 33.387 | 5.995 |
| After loss + `del logits` (pre-BW) | 33.004 | 37.293 | 2.088 |
| After BW | **37.760** | **38.191** | 6.969 |
| After `clip_grad_norm` | 37.760 | 38.191 | 6.969 |
| After `optimizer.step()` | 37.760 | 38.191 | 6.969 |
| *B=9 epoch-level peak (over all 15 steps)* | | **38.191** | **1.190** |
| *B=10, Epoch 0, step 0 (partial — OOM during BW)* | | | |
| Step start (peak reset) | 0.002 | 4.164 | 35.217 |
| After FW | 34.292 | 36.609 | 2.772 |
| After loss + `del logits` (pre-BW) | **36.670** | **38.094** | 2.676 |
| $\hookrightarrow$ *`loss.backward()` OOM: alloc = 36.55 GiB, free = 1.57 GiB, tried 4.89 GiB* | | | |

### 4.5.5 Key Observations

**1. Batch size advantage and measured speedup.** SegProp supports $B{=}13$ vs. FSDP's maximum $B{=}9$ on the same hardware, a **44%** larger batch. Throughput: SegProp $106.4\,\text{tok/s}$ ($\ell_{\text{GH}}{=}4$) / $110.7\,\text{tok/s}$ ($\ell_{\text{GH}}{=}2$) vs. FSDP $85.5\,\text{tok/s}$. Measured training times confirm this: SegProp completes 1 effective epoch in $2503.4\,\text{s}$ ($\ell_{\text{GH}}{=}4$; $\ell_{\text{GH}}{=}2$ takes $2405.9\,\text{s}$); FSDP completes 1 epoch ($15\,\text{steps} \times 9\,\text{samples} = 135$ samples) in $3231.9\,\text{s}$. SegProp's 3 sub-epochs correspond to one effective body update epoch; reported times are normalized by samples per effective epoch. Adjusted speedup, normalised by samples per epoch ($n = B \times \lceil 128/B \rceil$):

$$\text{speedup} = \frac{t_{\text{FSDP}}/n_{\text{FSDP}}}{t_{\text{SegProp}}/n_{\text{SegProp}}} = \frac{3231.9/135}{2503.4/130} = \frac{23.94}{19.26} \approx \mathbf{1.24\times} \tag{4.3}$$

**2. Why SegProp fits a larger batch.** SegProp's backward pass processes only $\ell_{\text{grad}} = 26{+}2 = 28$ layers (for $\ell_{\text{GH}}{=}2$) or $\approx 25{+}4 = 29$ layers (for $\ell_{\text{GH}}{=}4$), vs. FSDP's full $L{=}80$ layers. Activation checkpoint tensors scale with $\ell_{\text{grad}}$, not $L$, reducing backward-pass memory by $L/\ell_{\text{grad}} \approx 2.7\text{–}2.9\times$.

The core mechanism is *depth-wise gradient sparsity*: at any epoch, gradients and checkpointed backward state are required only for the active body segment and the global head, i.e., for $\ell_{\text{grad}} \ll L$ layers, while all earlier segments run forward-only. Consequently, the dominant checkpointed backward-pass activation memory scales as $O(\ell_{\text{grad}} \cdot B \cdot T \cdot d)$ rather than $O(L \cdot B \cdot T \cdot d)$, yielding an empirical $L/\ell_{\text{grad}} \approx 2.7\text{–}2.9\times$ reduction in this setting and enabling $B = 13$ where FSDP+offload is limited to $B = 9$ on the same hardware.

Table 4: SegProp $B$=13, $\ell_{\text{GH}}$=4. Peak reserved = 38.268 GiB (1.11 GiB headroom); peak alloc = 38.016 GiB (during backward, epoch 0, measured via reset peak tracker). $B$=14 OOMs during the backward pass of epoch 0 due to *allocated* memory exceeding the GPU total.

| Epoch | Checkpoint | peak_alloc (GiB) | peak_reserved (GiB) |
|---|---|---:|---:|
| *Epoch 0: Seg0 training* | | | |
| 0 | Step start | 6.466 | 9.709 |
| 0 | After seg0 fwd | 23.710 | 36.820 |
| 0 | After GH fwd | 25.335 | 38.246 |
| 0 | Backward **(peak alloc, peak reserved)** | **38.016** | **38.268** |
| 0 | After backward | 38.016 | 38.268 |
| 0 | After optimizer step | 38.016 | 38.268 |
| *Epoch 1: Seg0 frozen, Seg1 training* | | | |
| 1 | Step start | 18.742 | 19.381 |
| 1 | After frozen seg0 fwd | 24.771 | 28.463 |
| 1 | After seg0 offload $(-5.180\,\text{GiB})$ | 24.771 | 28.463 |
| 1 | After seg1 fwd | 24.771 | 38.189 |
| 1 | After GH fwd | 25.388 | 38.189 |
| 1 | Backward **(peak alloc)** | **37.817** | 38.150 |
| 1 | After backward | 37.817 | 38.150 |
| 1 | After optimizer step | 37.817 | 38.150 |
| *Epoch 2: Seg0+Seg1 frozen, Seg2 training* | | | |
| 2 | Step start | 18.100 | 18.775 |
| 2 | After seg0 reload $(+5.180\,\text{GiB})$ | 23.279 | 25.943 |
| 2 | After frozen seg0 fwd | 29.951 | 33.033 |
| 2 | After seg0 offload $(-5.180\,\text{GiB})$ | 29.951 | 33.033 |
| 2 | After frozen seg1 fwd | 29.951 | 33.033 |
| 2 | After seg1 offload $(-4.981\,\text{GiB})$ | 29.951 | 33.033 |
| 2 | After seg2 fwd | 29.951 | 38.229 |
| 2 | After GH fwd | 29.951 | 38.229 |
| 2 | Backward **(peak alloc)** | **37.817** | 38.229 |
| 2 | After backward | 37.817 | 38.229 |
| 2 | After optimizer step | 37.817 | 38.229 |

## 4.6 FLOPs Analysis: FSDP vs. SegProp for LLaMA-70B

This section compares the floating-point operations (FLOPs) per training sequence for standard FSDP and SegProp on LLaMA-70B.

### 4.6.1 Notation and Model Parameters

| Symbol | Value | Description |
|--------|-------|-------------|
| $L$ | 80 | Total decoder layers |
| $\ell_0$ | 26 | Layers in body segment 0 |
| $\ell_1$ | 25 | Layers in body segment 1 |
| $\ell_2$ | 25 | Layers in body segment 2 |
| $\ell_{\mathrm{GH}}$ | 4 | Decoder layers in Global Head |
| $d$ | 8,192 | Hidden size |
| $d_{\mathrm{ff}}$ | 28,672 | MLP intermediate size |
| $H$ | 64 | Query attention heads |
| $H_{\mathrm{kv}}$ | 8 | Key/value heads (GQA) |
| $d_h$ | 128 | Head dimension ($d/H$) |
| $V$ | 128,256 | Vocabulary size |
| $T$ | 2,048 | Sequence length (tokens) |

All FLOPs counts follow the standard convention of counting one multiply-add as 2 FLOPs. All values below are per training sequence. The **embedding layer** (`embed_tokens`) is a lookup table (index/gather operation) with no multiply-adds, so it contributes negligible FLOPs and is omitted. The **LM head** is a full linear projection ($\mathbf{W}_{\mathrm{lm}} \in \mathbb{R}^{V \times d}$) and is included explicitly. RMSNorm layers are also omitted as their FLOPs are negligible relative to attention and MLP.

### 4.6.2 Common FLOPs per Sequence

**Single decoder layer.** For one sequence of length $T$, the Grouped Query Attention (GQA) sub-layer contributes:

$$\mathcal{F}_{\mathrm{attn}} = 4Td(d + H_{\mathrm{kv}}d_h) + 4HT^2 d_h,$$

and the SwiGLU MLP sub-layer contributes:

$$\mathcal{F}_{\mathrm{mlp}} = 6 \cdot T \cdot d \cdot d_{\mathrm{ff}}.$$

Substituting LLaMA-70B values ($T$=2048, $d$=8192, $H_{\mathrm{kv}}d_h$=1024, $H$=64, $d_{\mathrm{ff}}$=28672):

$$\mathcal{F}_{\mathrm{attn}} \approx 756 \text{ GFLOPs}, \quad \mathcal{F}_{\mathrm{mlp}} \approx 2{,}886 \text{ GFLOPs},$$
$$\mathcal{F}_{\mathrm{layer}}^{\mathrm{fwd}} = \mathcal{F}_{\mathrm{attn}} + \mathcal{F}_{\mathrm{mlp}} \approx 3{,}642 \text{ GFLOPs} \approx 3.642 \text{ TFLOPs}.$$

**LM head.**

$$\mathcal{F}_{\mathrm{LM}}^{\mathrm{fwd}} = 2 \cdot T \cdot d \cdot V \approx 4{,}304 \text{ GFLOPs} \approx 4.304 \text{ TFLOPs}.$$

**Backward pass and activation checkpointing.** With activation checkpointing, the total FLOPs for a forward + backward pass are 4× the forward FLOPs: (i) 1× forward, (ii) 1× recompute, (iii) 2× backward.

$$\mathcal{F}_{\mathrm{layer}}^{\mathrm{total}} = 4 \cdot \mathcal{F}_{\mathrm{layer}}^{\mathrm{fwd}} \approx 14.57 \text{ TFLOPs}, \quad \mathcal{F}_{\mathrm{LM}}^{\mathrm{total}} = 4 \cdot \mathcal{F}_{\mathrm{LM}}^{\mathrm{fwd}} \approx 17.22 \text{ TFLOPs}.$$

For a *frozen* layer (forward only):

$$\mathcal{F}_{\mathrm{layer}}^{\mathrm{frozen}} = \mathcal{F}_{\mathrm{layer}}^{\mathrm{fwd}} \approx 3.642 \text{ TFLOPs}.$$

### 4.6.3 FSDP FLOPs per Sequence

All $L$=80 layers backpropagate at every step:

$$\mathcal{F}_{\mathrm{FSDP}} = L \cdot \mathcal{F}_{\mathrm{layer}}^{\mathrm{total}} + \mathcal{F}_{\mathrm{LM}}^{\mathrm{total}} = 80 \times 14.57 + 17.22 \approx 1{,}183 \text{ TFLOPs/seq.}$$

### 4.6.4 SegProp FLOPs per Sequence

The 80 decoder layers are partitioned into three body segments and one Global Head (GH): $\ell_0 = 26$, $\ell_1 = 25$, $\ell_2 = 25$, $\ell_{\mathrm{GH}} = 4$. At each stage, one body segment and the GH are trained (fwd+bwd+recompute); all previously trained segments are frozen (forward only).

**Stage 0: seg0 + GH active, no frozen segments.**   Active layers: $\ell_0 + \ell_{\mathrm{GH}} = 30$.

$$\mathcal{F}_{\mathrm{SP}}^{(0)} = 30 \times 14.57 + 17.22 \approx 454 \text{ TFLOPs/seq.}$$

**Stage 1: seg1 + GH active, seg0 frozen.**   Active: $\ell_1 + \ell_{\mathrm{GH}} = 29$. Frozen: $\ell_0 = 26$.

$$\mathcal{F}_{\mathrm{SP}}^{(1)} = \ell_0 \cdot \mathcal{F}_{\mathrm{layer}}^{\mathrm{frozen}} + 29 \times 14.57 + 17.22 \approx 95 + 423 + 17 \approx 535 \text{ TFLOPs/seq.}$$

**Stage 2: seg2 + GH active, seg0 and seg1 frozen.**   Active: $\ell_2 + \ell_{\mathrm{GH}} = 29$. Frozen: $\ell_0 + \ell_1 = 51$.

$$\mathcal{F}_{\mathrm{SP}}^{(2)} = (\ell_0 + \ell_1) \cdot \mathcal{F}_{\mathrm{layer}}^{\mathrm{frozen}} + 29 \times 14.57 + 17.22 \approx 186 + 423 + 17 \approx 626 \text{ TFLOPs/seq.}$$

### 4.6.5 Comparison

Table 5: FLOPs per training sequence: FSDP vs. SegProp for LLaMA-70B. "Active" layers undergo fwd+bwd+recompute ($4\times$ fwd); "frozen" layers undergo fwd only ($1\times$ fwd). The SegProp total sums all three stages, constituting one effective training pass through all body layers.

| Method | Active layers | Frozen layers | FLOPs/seq (TFLOPs) | vs. FSDP |
|---|---|---|---|---|
| FSDP (1 pass) | 80 | 0 | 1,183 | $1.00\times$ |
| SegProp stage 0 | 30 | 0 | 454 | $0.38\times$ |
| SegProp stage 1 | 29 | 26 | 535 | $0.45\times$ |
| SegProp stage 2 | 29 | 51 | 626 | $0.53\times$ |
| **SegProp total (3 stages)** | **80** | — | **1,615** | **$1.37\times$** |

As seen in Table 5, SegProp uses $1.37\times$ FSDP's per-sequence FLOPs (1,615 vs. 1,183 TFLOPs/seq). The overhead arises from frozen-segment forward passes in stages 1 and 2, plus several traversals of the global head. Frozen segments contribute forward-only FLOPs ($\frac{1}{4}$ the cost of an active layer). SegProp accepts a 37% total FLOPs overhead in exchange for a reduced per-stage backward pass.

### 4.7 SegProp Fine-Tuning for LLMs

Figure 5 compares Nemo MMLU (5-shot) across depth for SegProp—shown with two global heads (lm-head only and lm-head + 2 layers)—and layer-wise fine-tuning (LW).

Across base depths, SegProp consistently outperforms LW on Nemo MMLU (5-shot) (Figure 5). For a small base (1 layer), both SegProp variants quickly rise through later layers and converge near the top, while LW remains low across the stack. With deeper bases (18 and 25 layers), SegProp rapidly approaches its final performance shortly after the base boundary, and the 'lm-head + 2 layers' variant provides a small but consistent gain over lm-head-only. Overall, maintaining a global head during segmented fine-tuning consistently improves intermediate-layer performance across the evaluated bases, and modestly extending the global head yields further gains.

Parameter-efficient fine-tuning (PEFT) approaches such as LoRA are highly effective for LLM adaptation under constrained compute and memory budgets. SegProp is not positioned as a replacement for PEFT, since the two techniques address different axes of the training trade-off. In practice, SegProp can be composed with PEFT by applying LoRA within the trainable segment and/or within the persistent global head, if

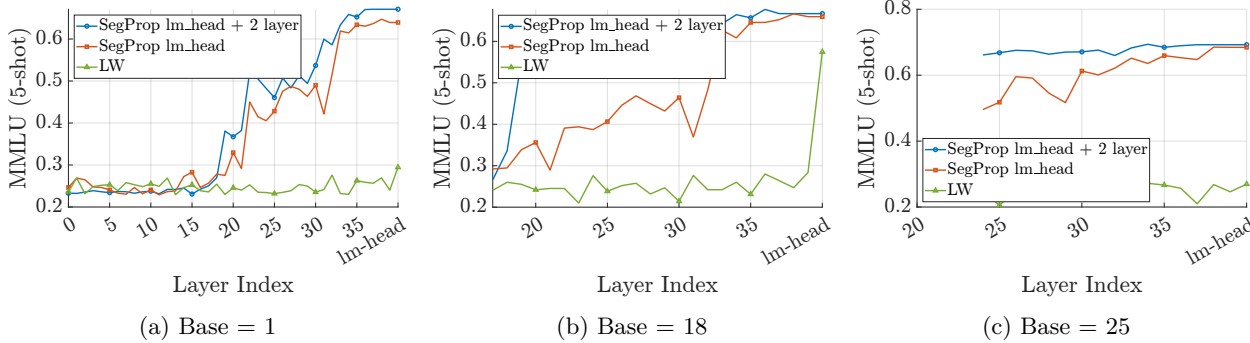

(a) Base = 1          (b) Base = 18          (c) Base = 25

Figure 5: Nemo MMLU (5-shot) versus layer index during segmented fine-tuning for SegProp (two variants: lm-head only and lm-head + 2 layers) and layer-wise fine-tuning (LW) across three base prefixes. The final x-axis tick corresponds to the lm-head.

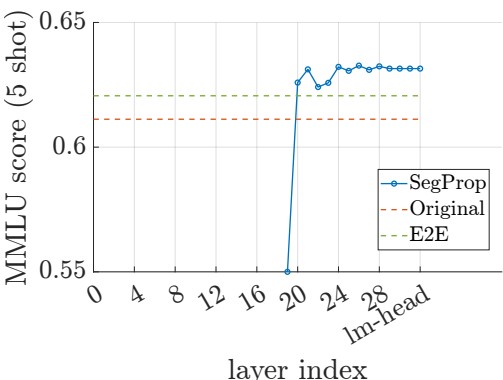
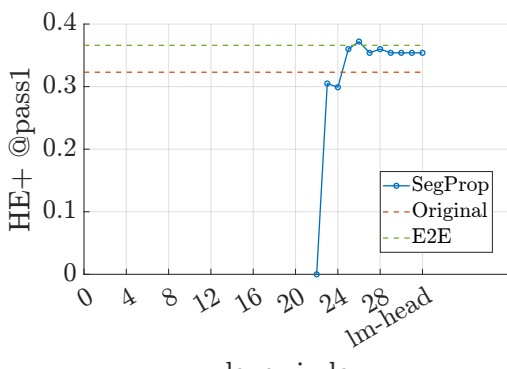

(a) MMLU (5-shot) across layers; base = 21. Comparison between SegProp (lm_head + 2 layers), Original, and E2E fine-tune.

(b) HE+ @pass1 across layers; base = 24. Comparison between SegProp (lm_head + 2 layers), Original, and E2E fine-tune.

Figure 6: Segmented fine-tuning diagnostics for Mistral-7B at two bases. Left: Base-21 MMLU (5-shot) versus layer index. Right: Base-24 coding HE+ @pass1 versus layer index. In both panels, SegProp uses a persistent global head extended by two layers (lm_head + 2) and is compared against the out-of-the-box model (Original) and an end-to-end (E2E) fine-tuned model. The final x-axis tick corresponds to the lm-head.

desired. This subsection is included primarily to provide empirical evidence on segmented fine-tuning across base depths and to illustrate how a persistent global head affects intermediate-layer behavior.

Figure 6 shows Mistral-7B segmented fine-tuning with a short persistent global head (lm_head + 2 layers). For MMLU (base = 21), SegProp scores increase sharply just beyond the base boundary, surpassing the original model and meeting the E2E fine-tuned model near the top layers. For coding (HE+, base = 24), SegProp again rises quickly at the boundary and matches the E2E baseline while remaining clearly above the original. These trends indicate that keeping a short global head active maintains a consistent end-task training signal across stages and allows segmented fine-tuning to nearly match full E2E fine-tuning in these settings.

As summarized in Table 6, segmented fine-tuning with an appropriately chosen base depth achieves performance that meets end-to-end (E2E) fine-tuning on both Mistral-Nemo-Instruct-2407 and Llama3.1-8B-Instruct. For Mistral-Nemo, all evaluated bases remain close to the E2E Winogrande score, and the deeper bases effectively match it. For Llama3.1-8B-Instruct, bases 18, 21, and 24 reach MMLU and Winogrande accuracy that meets the E2E fine-tune. These results confirm that SegProp can deliver E2E-level fine-tune quality without updating the entire network and are consistent with the trends observed in Figures 5 and 6.

Table 6: Segmented fine-tuning performance across base depths for Mistral-Nemo-Instruct-2407 and Llama3.1-8B-Instruct on MMLU and Winogrande (5-shot).

| Model | Base | MMLU (5-shot) | Winogrande (5-shot) |
|---|---|---|---|
| Mistral-Nemo-Instruct-2407 | out of the box | 67.70% | 82.79% |
| | E2E Fine-Tune | 69.31% | 82.72% |
| | 1 | 67.09% | 81.29% |
| | 18 | 66.70% | 80.90% |
| | 25 | 69.38% | 81.37% |
| | 32 | 68.46% | 82.24% |
| Llama3.1-8B-Instruct | out of the box | 68.20% | 77.90% |
| | E2E Fine-Tune | 68.39% | 79.08% |
| | 1 | 67.53% | 77.19% |
| | 18 | 68.22% | 78.45% |
| | 21 | 68.45% | 78.93% |
| | 24 | 68.12% | 78.45% |

## 5  Discussion and Future Work

Empirically, SegProp training accuracy approaches end-to-end (E2E) training accuracy. This holds across CNN backbones and ImageNet-scale ViT training, and extends to LLM regimes: SegProp matches or nearly matches E2E fine-tuning on 7–12B models, and our LLaMA-70B study shows that the same depth-wise gradient sparsity enables larger feasible batches under 40 GiB GPUs, with an explicit compute–memory trade-off quantified by our FLOPs analysis.

Conceptually, SegProp sits between fully local learning and monolithic backpropagation. Like deep supervision (Belilovsky et al., 2018; Marquez et al., 2018), it injects task-level signal at intermediate depths, but it does so via the *actual* final layers rather than many separate auxiliary heads. This avoids proliferating objectives and sidesteps inference-time discrepancies, since the global head used during training is identical to the head used at test time.

SegProp is also complementary to existing memory-efficiency techniques. Its segmented optimization naturally reduces the depth over which gradients must be stored or propagated, making it compatible with activation checkpointing and selective recomputation (Chen et al., 2016; Korthikanti et al., 2022). Snapshot checkpointing (SnapCheck) can further reduce compute by caching detached prefix activations once a prefix is frozen and reusing them in later stages, which is especially attractive in multi-GPU and resource-constrained settings. Combining SegProp with parameter-efficient fine-tuning methods such as LoRA—for example, using LoRA within selected segments or within the persistent head—could further improve the trade-off between accuracy, memory, and wall-clock cost, especially in low-budget fine-tuning regimes.

We do not measure HSIC or mutual information directly; we cite HSIC/IB only as qualitative motivation and focus on empirical accuracy–memory–compute trade-offs.

**Limitations.**  Our experiments cover CNN backbones, ImageNet-scale ViT training, and a system-level feasibility and FLOPs analysis for LLaMA-70B under a specific hardware/sharding configuration. We do not study extremely deep transformers, mixture-of-experts or multimodal architectures, or large-scale pre-training from scratch, where dynamics and system constraints may differ. We also focus on SegProp-style segmented fine-tuning and do not provide a systematic comparison to popular parameter-efficient methods such as LoRA, which offer an alternative way to reduce trainable parameters and memory footprint. Our training-only adapters are manually designed rather than learned, and we provide qualitative but not formal guarantees on when SegProp fully recovers E2E behavior.

Looking ahead, applying SegProp to larger and more heterogeneous models (e.g., MoE, multimodal, or hybrid CNN–transformer stacks) is a natural next step. Automatically designing adapters, stage schedules,

or freezing policies may sharpen these trade-offs further. A more formal information-theoretic analysis, building on HSIC and information-bottleneck tools, may clarify when a single persistent head is sufficient and when richer global supervision pathways are beneficial. Overall, our findings support a simple design principle: *segmentation is not inherently at odds with global supervision*. A small, expressive global head can recover much of the empirical performance of E2E training in the evaluated settings, while enabling more flexible and efficient segmented optimization.

Our LLaMA-70B results are a feasibility and compute–memory trade-off study (max feasible batch size, measured memory traces, FLOPs), and do not claim improved pre-training quality (e.g., perplexity) relative to end-to-end training.

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

# A  SegProp-SGD Algorithm Details

---

**Algorithm 1** Segmented Propagation Stochastic Gradient Descent (SegProp-SGD)

---

**Inputs:** Network $f(x) = h \circ f_{n-1} \circ f_{n-2} \circ \cdots \circ f_0(x)$ with $n$ blocks; base depth $p$; last-layers size $r$; loss $\ell$; optimizer(s) $\mathcal{O}$; exit criterion for Stage 1; exit criterion for block $\hat{p}$; global exit criterion

1: **Notation:** $F_{a:b}(x) := f_{b-1} \circ f_{b-2} \circ \cdots \circ f_a(x);$ $f^{[0:p]}(x) := F_{0:p}(x),\ p \in \{0, \ldots, n-2\};$ $\mathcal{LL}^{(r)} :=$ $h \circ F_{n-r:n},\ r \in \{1, \ldots, n-p\};$ optional adapters $A_j$ (identity if unused)

2: **Partition:** Decompose $f$ into non-overlapping blocks $f_0, \ldots, f_{n-1}$ (e.g., ResNet stages or Transformer layers). Always include $\mathcal{LL}^{(r)}$ during training to provide global supervision.

 

*Stage 1: Joint training of base prefix with $\mathcal{LL}^{(r)}$*

3: **while** Stage 1 exit criterion not met **do**

4:     **for** each minibatch $(x, y)$ **do**

5:         $z \leftarrow f^{[0:p]}(x)$

6:         $\tilde{z} \leftarrow A_p(z)$                    ▷ Adapter for geometry matching; identity if not needed

7:         $\hat{y} \leftarrow \mathcal{LL}^{(r)}(\tilde{z})$

8:         $\mathcal{L}_1 \leftarrow \ell(\hat{y}, y)$

9:         **Backward/Update:** backprop through $\{f^{[0:p]}, \mathcal{LL}^{(r)}, A_p\}$ and update only these parameters

10: **Commit and freeze** $f^{[0:p]}$; keep $\mathcal{LL}^{(r)}$ **trainable and persistent** for Stage 2

 

*Stage 2: Iterative training of intermediate blocks with persistent $\mathcal{LL}^{(r)}$*

11: Initialize snapshot cache $\mathcal{S}$ as empty

12: **for** $\hat{p} \leftarrow p$ **to** $n-r-1$ **do**

13:     **while** exit criterion for block $\hat{p}$ not met **do**

14:         **for** each minibatch $(x, y)$ with identifier batch_id **do**

15:             **if** snapshot $\mathcal{S}[p, \text{batch\_id}]$ exists **then**

16:                 $z \leftarrow \mathcal{S}[p, \text{batch\_id}]$

17:             **else**

18:                 $z \leftarrow \text{StopGrad}\big(f^{[0:p]}(x)\big)$

19:                 Optionally store $\mathcal{S}[p, \text{batch\_id}] \leftarrow z$        ▷ SnapCheck: cache activations

20:             $z_{\hat{p}} \leftarrow f_{\hat{p}}(z)$

21:             $\tilde{z}_{\hat{p}} \leftarrow A_{\hat{p}}(z_{\hat{p}})$                 ▷ Adapter; identity if not needed

22:             $\hat{y} \leftarrow \mathcal{LL}^{(r)}(\tilde{z}_{\hat{p}})$

23:             $\mathcal{L}_2 \leftarrow \ell(\hat{y}, y)$

24:             **Backward/Update:** backprop through $\{f_{\hat{p}}, \mathcal{LL}^{(r)}, A_{\hat{p}}\}$; update only $\{f_{\hat{p}}, \mathcal{LL}^{(r)}, A_{\hat{p}}\}$

25:     **Commit and freeze** $f_{\hat{p}}$; **discard** $A_{\hat{p}}$; **extend base** $p \leftarrow \hat{p}$

26:     Optionally clear or rebuild $\mathcal{S}$ for the new prefix depth $p$

27:     **if** global exit criterion satisfied **then break**

28: **Output:** committed backbone $F_{0:n}$ and $\mathcal{LL}^{(r)}$; inference uses the standard forward $h \circ F_{0:n}$ (adapters and snapshots discarded)

---

# B  Architectures and Training Regimes: ResNet-18 and ResNet-50

## B.1  Common Data & Optimization Settings

**Dataset and preprocessing.** All experiments use CIFAR-10 with ImageNet-style crops for stronger invariance:

- **Train transforms:** `Resize(256)` $\rightarrow$ `RandomResizedCrop(224, scale=(0.6,1.0))` $\rightarrow$ `RandomHorizontalFlip()` $\rightarrow$ `RandAugment(num_ops=2, magnitude=9)` $\rightarrow$ `ToTensor()`

```
→        Normalize(mean=(0.4914,0.4822,0.4465), std=(0.2023,0.1994,0.2010))        →
RandomErasing(p=0.25, scale=(0.02,0.2), ratio=(0.3,3.3)).
```

- **Test transforms:** `Resize(256)` → `CenterCrop(224)` → `ToTensor()` → `Normalize` (same statistics).

**Optimization.** Unless specified otherwise, we use **AdamW** with lr $= 3 \times 10^{-4}$, $\beta_1 = 0.9$, $\beta_2 = 0.999$, weight decay $= 0.01$, batch size $= 256$, mixed precision (AMP), cross-entropy loss. E2E uses 400 epochs; LW/SegProp use 400 epochs per stage.

### B.2 ResNet-18

### B.2.1 Backbone Segmentation and Head Definitions

**Backbone split (224×224 inputs).**

**seg1. seg1:** conv1 → bn1 → ReLU → maxpool → layer1 produces $[64, 56, 56]$.

**seg2. seg2:** layer2 produces $[128, 28, 28]$.

**seg3. seg3:** layer3 produces $[256, 14, 14]$.

**seg4. seg4:** layer4[0] (full) + layer4[1].(conv1 → bn1 → ReLU), producing identity $\mathbf{I} \in \mathbb{R}^{B \times 512 \times 7 \times 7}$ and pre-activation $\mathbf{P} \in \mathbb{R}^{B \times 512 \times 7 \times 7}$.

**Global head (SegProp).** Owns *only* layer4[1].conv2 and bn2 (last conv + BN), then:

$$\hat{\mathbf{Z}} = \mathrm{bn2}(\mathrm{conv2}(\mathbf{P})) + \mathbf{I}, \qquad \hat{\mathbf{Z}} \leftarrow \mathrm{ReLU}(\hat{\mathbf{Z}}).$$

Global average pooling and a two-layer MLP ($512 \to H \to 10$, $H = 512$, dropout 0.3) produce the logits.

### B.2.2 E2E (End-to-End) Training

- **Structure:** Standard ResNet-18, classifier replaced by MLP $512 \to 512 \to 10$.

- **Flow:** Full network forward; full network backprop each epoch.

- **Schedule:** 400 epochs; AdamW (as above); AMP on.

### B.2.3 LW (Layer-Wise) Training

**Auxiliaries.** Each stage uses a **Conv–BN–ReLU mapping adapter** to normalize to the canonical head resolution, followed by **GAP + Linear** for local classification:

- Stage 1 aux: $[64, 56, 56] \to [512, 7, 7]$ via strided convs $\Rightarrow$ GAP $\Rightarrow$ Linear($512 \to 10$).

- Stage 2 aux: $[128, 28, 28] \to [512, 7, 7] \Rightarrow$ GAP $\Rightarrow$ Linear.

- Stage 3 aux: $[256, 14, 14] \to [512, 7, 7] \Rightarrow$ GAP $\Rightarrow$ Linear.

- Stage 4 aux: input $[512, 7, 7]$ (no mapping) $\Rightarrow$ GAP $\Rightarrow$ Linear.

We use 5 LW stages for ResNet-18: **seg1**, **seg2**, **seg3**, **seg4** (each with an auxiliary head), and a final **fc** stage that trains only the global classifier with the backbone frozen.

**Flow and sizes.**

| Stage | Forward (to aux) | Aux input size |
|---|---|---|
| 1 | $\text{seg1}(x)$ | $[64, 56, 56]$ |
| 2 | $\text{seg1} \to \text{seg2}$ | $[128, 28, 28]$ |
| 3 | $\text{seg1} \to \text{seg2} \to \text{seg3}$ | $[256, 14, 14]$ |
| 4 | $\text{seg1} \to \text{seg2} \to \text{seg3} \to \text{seg4}$ | $[512, 7, 7]$ |

**Freezing.** Previous segments are frozen; only the current segment + aux are trainable.

### B.2.4 SegProp Training

**Auxiliaries (mapping only).** Aux1/Aux2/Aux3 output *both* $(\mathbf{I}, \mathbf{P})$ in $\mathbb{R}^{B \times 512 \times 7 \times 7}$:

- Aux1: from $[64, 56, 56]$ to $(\mathbf{I}, \mathbf{P})$.

- Aux2: from $[128, 28, 28]$ to $(\mathbf{I}, \mathbf{P})$.

- Aux3: from $[256, 14, 14]$ to $(\mathbf{I}, \mathbf{P})$.

**Flow and sizes.**

| Stage | Forward | Aux output | Head input |
|---|---|---|---|
| 1 | $\text{seg1} \xrightarrow{\text{aux1}}$ | $(\mathbf{I}, \mathbf{P})$ | $[512, 7, 7]$ each |
| 2 | $\text{seg1} \to \text{seg2} \xrightarrow{\text{aux2}}$ | $(\mathbf{I}, \mathbf{P})$ | $[512, 7, 7]$ each |
| 3 | $\text{seg1} \to \text{seg2} \to \text{seg3} \xrightarrow{\text{aux3}}$ | $(\mathbf{I}, \mathbf{P})$ | $[512, 7, 7]$ each |
| 4 | $\text{seg1} \to \text{seg2} \to \text{seg3} \to \text{seg4}$ | $(\mathbf{I}, \mathbf{P})$ | $[512, 7, 7]$ each |

**Freezing.** At each stage, previous segments are frozen. In Stages 1–3 the aux feeds the head directly. Stage 4 trains **seg4 + head**.

### B.3 ResNet-50

We adopt the standard ImageNet-style ResNet-50 (7×7 stride-2 stem and max pool; 224×224 inputs) and view the backbone as four segments $f_1, \ldots, f_4$ producing intermediate representations:

$$z_1 = f_1(x), \quad z_2 = f_2(z_1), \quad z_3 = f_3(z_2), \quad z_4 = f_4(z_3).$$

### B.3.1 Backbone Segmentation (Common to All Head Variants)

**Backbone split (224×224 inputs).**

**seg1. seg1** $(f_1)$**:** stem + layer1, output $z_1 \in \mathbb{R}^{B \times 256 \times 56 \times 56}$.

**seg2. seg2** $(f_2)$**:** layer2, output $z_2 \in \mathbb{R}^{B \times 512 \times 28 \times 28}$.

**seg3. seg3** $(f_3)$**:** layer3, output $z_3 \in \mathbb{R}^{B \times 1024 \times 14 \times 14}$.

**seg4. seg4 backbone:** the final residual stage layer4, decomposed differently depending on how many convolutions are moved into the global head. We always arrange it so that $f_4$ produces an identity tensor $\mathbf{I}$ from the residual branch and a "pre-head" tensor that serves as input to the global head.

Below we describe three SegProp instantiations that differ in how the last bottleneck layer4[2] is split between the backbone and the global head.

### B.3.2 ResNet-50 E2E Training

- **Structure:** Standard ResNet-50, replacing the ImageNet classifier with an MLP $2048 \rightarrow 512 \rightarrow 10$.

- **Flow:** Full network forward:

$$x \rightarrow f_1 \rightarrow f_2 \rightarrow f_3 \rightarrow f_4 \rightarrow \text{GAP} \rightarrow \text{MLP},$$

  and full network backprop each epoch.

- **Schedule:** 400 epochs; AdamW; AMP on.

### B.3.3 ResNet-50 LW Training

**Auxiliaries.** For LW (layer-wise) training, we attach local heads that operate on a canonical $[2048, 7, 7]$ representation. Each stage uses a Conv–BN–ReLU adapter to map the stage output to $[2048, 7, 7]$, followed by GAP + Linear($2048 \rightarrow 10$):

- Stage 1 aux: $[256, 56, 56] \rightarrow [2048, 7, 7] \rightarrow \text{GAP} \rightarrow \text{Linear}$.

- Stage 2 aux: $[512, 28, 28] \rightarrow [2048, 7, 7] \rightarrow \text{GAP} \rightarrow \text{Linear}$.

- Stage 3 aux: $[1024, 14, 14] \rightarrow [2048, 7, 7] \rightarrow \text{GAP} \rightarrow \text{Linear}$.

- Stage 4 aux: input $[2048, 7, 7] \rightarrow \text{GAP} \rightarrow \text{Linear}$.

**Freezing.** At stage $s$, we freeze $f_1, \ldots, f_{s-1}$ and train only $f_s$ and its local aux head.

### B.3.4 ResNet-50 SegProp: 1-Conv, 2-Conv, and 3-Conv Global Heads

SegProp augments the backbone with small auxiliary adapters $g_1, g_2, g_3$ that map intermediate features $z_1, z_2, z_3$ to the exact tensors expected by a *shared global head $H$*. At stage $s \in \{1, 2, 3\}$ we use $g_s$ to feed $H$ directly; at stage 4 we use the real backbone output. The difference between our three ResNet-50 SegProp variants lies in how many convolutions of the last bottleneck layer4[2] are assigned to $H$.

**Notation for the last bottleneck.** We denote the three conv+BN blocks of layer4[2] by

$$\text{conv1} + \text{bn1}, \qquad \text{conv2} + \text{bn2}, \qquad \text{conv3} + \text{bn3},$$

with standard ResNet-50 widths:

$$\text{conv1} : 1024 \rightarrow 512, \quad \text{conv2} : 512 \rightarrow 512, \quad \text{conv3} : 512 \rightarrow 2048.$$

**(A) 1-Conv global head (conv3+bn3 only).** This variant corresponds to the simplest split: the head owns only the final conv+BN of layer4[2]; the backbone owns the rest of layer4.

**Backbone seg4 (1-conv head).**

- seg4 includes: layer4[0], layer4[1], and all of layer4[2].(conv1 $\rightarrow$ bn1 $\rightarrow$ ReLU $\rightarrow$ conv2 $\rightarrow$ bn2 $\rightarrow$ ReLU).

- It produces:
$$\mathbf{I} \in \mathbb{R}^{B \times 2048 \times 7 \times 7}, \qquad \mathbf{P} \in \mathbb{R}^{B \times 512 \times 7 \times 7},$$

  where $\mathbf{I}$ is the residual identity (from the skip branch) and $\mathbf{P}$ is the pre-activation input to conv3.

**Global head $H^{(1)}$ (1-conv).** Given $(\mathbf{I}, \mathbf{P})$, the head applies the last conv+BN and the classifier:

$$\mathbf{V} = \text{bn3}(\text{conv3}(\mathbf{P})) \in \mathbb{R}^{B \times 2048 \times 7 \times 7},$$

$$\hat{\mathbf{Z}} = \text{ReLU}(\mathbf{V} + \mathbf{I}), \qquad \mathbf{h} = \text{GAP}(\hat{\mathbf{Z}}) \in \mathbb{R}^{B \times 2048},$$

$$\mathbf{o} = \mathbf{W}_2 \, \sigma(\mathbf{W}_1 \mathbf{h} + \mathbf{b}_1) + \mathbf{b}_2,$$

with MLP $2048 \to 512 \to 10$.

**SegProp auxiliaries for 1-conv head.** For stages $s \in \{1, 2, 3\}$, we define

$$x^{(1)} = z_1 \in \mathbb{R}^{B \times 256 \times 56 \times 56}, \quad x^{(2)} = z_2 \in \mathbb{R}^{B \times 512 \times 28 \times 28}, \quad x^{(3)} = z_3 \in \mathbb{R}^{B \times 1024 \times 14 \times 14}.$$

Each aux is a one-conv adapter:

$$h^{(s)} = \sigma\big(\text{BN}(\text{Conv}_{1\times 1}^{s7}(x^{(s)}))\big) \in \mathbb{R}^{B \times 2560 \times 7 \times 7},$$

$$\mathbf{I}_s = h^{(s)}[:, 1{:}2048, :, :], \qquad \mathbf{P}_s = h^{(s)}[:, 2049{:}2560, :, :],$$

with strides $s_7 \in \{8, 4, 2\}$ for seg1/2/3 to reach $7 \times 7$. These $(\mathbf{I}_s, \mathbf{P}_s)$ are fed into $H^{(1)}$.

**(B) 2-Conv global head (conv2+bn2 + conv3+bn3).** This variant moves the last two conv blocks of layer4[2] into the head, leaving only conv1+bn1 in the backbone.

**Backbone seg4 (2-conv head).**

- seg4 includes: layer4[0], layer4[1], and layer4[2].(conv1 $\to$ bn1 $\to$ ReLU).

- It produces:
$$\mathbf{I} \in \mathbb{R}^{B \times 2048 \times 7 \times 7}, \qquad \mathbf{P} \in \mathbb{R}^{B \times 512 \times 7 \times 7},$$
where $\mathbf{P}$ is now the output of conv1 + bn1 + ReLU of the last bottleneck.

**Global head $H^{(2)}$ (2-conv).** Given $(\mathbf{I}, \mathbf{P})$, the head applies conv2+bn2 and conv3+bn3, followed by the classifier:

$$\mathbf{U} = \text{ReLU}\big(\text{bn2}(\text{conv2}(\mathbf{P}))\big) \in \mathbb{R}^{B \times 512 \times 7 \times 7},$$

$$\mathbf{V} = \text{bn3}(\text{conv3}(\mathbf{U})) \in \mathbb{R}^{B \times 2048 \times 7 \times 7},$$

$$\hat{\mathbf{Z}} = \text{ReLU}(\mathbf{V} + \mathbf{I}), \qquad \mathbf{h} = \text{GAP}(\hat{\mathbf{Z}}) \in \mathbb{R}^{B \times 2048},$$

$$\mathbf{o} = \mathbf{W}_2 \, \sigma(\mathbf{W}_1 \mathbf{h} + \mathbf{b}_1) + \mathbf{b}_2.$$

**SegProp auxiliaries for 2-conv head.** The aux design is identical to the 1-conv case: each $g_s$ maps $x^{(s)}$ to $(\mathbf{I}_s, \mathbf{P}_s)$ with shapes $[2048, 7, 7]$ and $[512, 7, 7]$ via a single $1 \times 1$ conv producing 2560 channels (2048+512) at $7 \times 7$:

$$h^{(s)} = \sigma\big(\text{BN}(\text{Conv}_{1\times 1}^{s7}(x^{(s)}))\big) \in \mathbb{R}^{B \times 2560 \times 7 \times 7},$$

$$\mathbf{I}_s = h^{(s)}[:, 1{:}2048, :, :], \qquad \mathbf{P}_s = h^{(s)}[:, 2049{:}2560, :, :].$$

These are fed into $H^{(2)}$.

**(C) 3-Conv global head (conv1+bn1 + conv2+bn2 + conv3+bn3).** In our most "head-heavy" variant, the global head owns *all three* conv blocks of the last bottleneck layer4[2]. The backbone seg4 now includes only layer4[0] and layer4[1] (plus the residual path up to the point where $\mathbf{I}$ is defined).

**Backbone seg4 (3-conv head).**

- seg4 includes: layer4[0] and layer4[1] (full bottlenecks).

- It produces:
$$\mathbf{I} \in \mathbb{R}^{B \times 2048 \times 7 \times 7}, \qquad \mathbf{F}_{\text{in}} \in \mathbb{R}^{B \times 1024 \times 7 \times 7},$$
    where $\mathbf{F}_{\text{in}}$ is the input to layer4[2].conv1.

**Global head $H^{(3)}$ (3-conv).** Given $(\mathbf{I}, \mathbf{F}_{\text{in}})$, the head applies all three conv blocks of layer4[2] and the classifier:
$$\mathbf{U}_1 = \text{ReLU}\big(\text{bn1}(\text{conv1}(\mathbf{F}_{\text{in}}))\big) \in \mathbb{R}^{B \times 512 \times 7 \times 7},$$
$$\mathbf{U}_2 = \text{ReLU}\big(\text{bn2}(\text{conv2}(\mathbf{U}_1))\big) \in \mathbb{R}^{B \times 512 \times 7 \times 7},$$
$$\mathbf{V} = \text{bn3}(\text{conv3}(\mathbf{U}_2)) \in \mathbb{R}^{B \times 2048 \times 7 \times 7},$$
$$\hat{\mathbf{Z}} = \text{ReLU}(\mathbf{V} + \mathbf{I}), \qquad \mathbf{h} = \text{GAP}(\hat{\mathbf{Z}}) \in \mathbb{R}^{B \times 2048},$$
$$\mathbf{o} = \mathbf{W}_2 \, \sigma(\mathbf{W}_1 \mathbf{h} + \mathbf{b}_1) + \mathbf{b}_2.$$

**SegProp auxiliaries for 3-conv head.** Now the head input is $(\mathbf{I}, \mathbf{F}_{\text{in}})$ with shapes $[2048, 7, 7]$ and $[1024, 7, 7]$. For stages $s \in \{1, 2, 3\}$ we use one-conv adapters that output $2048 + 1024 = 3072$ channels at $7 \times 7$:
$$h^{(s)} = \sigma\big(\text{BN}(\text{Conv}_{1 \times 1}^{s7}(x^{(s)}))\big) \in \mathbb{R}^{B \times 3072 \times 7 \times 7},$$
$$\mathbf{I}_s = h^{(s)}[:, 1{:}2048, :, :], \qquad \mathbf{F}_{\text{in}}^{(s)} = h^{(s)}[:, 2049{:}3072, :, :],$$
with strides $s_7 \in \{8, 4, 2\}$ for seg1/2/3. These $(\mathbf{I}_s, \mathbf{F}_{\text{in}}^{(s)})$ are fed into $H^{(3)}$.

**Stage-wise SegProp schedule (all head variants).** For any of the three heads $H^{(k)}$ ($k \in \{1, 2, 3\}$), the SegProp schedule is:

- **Stage 1:** forward $x \to f_1 \to x^{(1)}$, apply aux $g_1$ to get the appropriate pair $(\mathbf{I}_1, \cdot)$, feed to $H^{(k)}$, and optimize $\mathcal{L}(\mathbf{o}^{(1)}, y)$. Trainable: $f_1$, $g_1$, $H^{(k)}$; frozen: $f_2, f_3, f_4$ and other aux.

- **Stage 2:** forward $x \to f_1 \to f_2 \to x^{(2)}$, apply $g_2$, feed to $H^{(k)}$. Trainable: $f_2$, $g_2$, $H^{(k)}$; frozen: $f_1, f_3, f_4$.

- **Stage 3:** forward $x \to f_1 \to f_2 \to f_3 \to x^{(3)}$, apply $g_3$, feed to $H^{(k)}$. Trainable: $f_3$, $g_3$, $H^{(k)}$; frozen: $f_1, f_2, f_4$.

- **Stage 4:** forward along the real backbone, including the appropriate seg4 for that head variant, to obtain the real pair (e.g., $(\mathbf{I}, \mathbf{P})$ or $(\mathbf{I}, \mathbf{F}_{\text{in}})$), feed to $H^{(k)}$, and optimize $\mathcal{L}(\mathbf{o}^{(4)}, y)$. Trainable: $f_4$ and $H^{(k)}$; frozen: $f_1, f_2, f_3$ and all aux.

At all stages, the global head $H^{(k)}$ is shared and updated, while each backbone segment is trained only in its own stage. Auxiliary modules are training-only (Stages 1–3) and are discarded after training.

**Modes and stability.** Frozen modules are set to `eval()` so BatchNorm running statistics do not update; only the current-stage segment, its aux, and the head are in `train()` mode. We use mixed precision (AMP) with unscale→clip→step to avoid occasional FP16 spikes.

## B.4 Schedules and Hyperparameters (Summary)

**E2E.** 400 epochs, AdamW ($3 \times 10^{-4}$, $\beta = (0.9, 0.999)$, wd $= 0.01$), batch 256, AMP on. Classifier heads:

- ResNet-18: MLP $512 \rightarrow 512 \rightarrow 10$.

- ResNet-50: MLP $2048 \rightarrow 512 \rightarrow 10$.

**LW.** 5 stages (seg1..seg4 + fc), ~400 epochs per stage; per-stage: *current segment + aux trainable; previous frozen*; aux = **Conv–BN–ReLU adapters** to canonical resolution $\rightarrow$ GAP + Linear; we report *Test (aux)* per stage and *Test (real)* through the assembled model.

**SegProp.** 4 stages (seg1..seg4); global head *always active*. Stages 1–3: do not execute the next segments beyond the current stage; aux produces the exact head inputs (ResNet-18: $(\mathbf{I}, \mathbf{P})$ with 512 channels; ResNet-50: $(\mathbf{I}, \mathbf{F}_{\text{in}})$ with 2048 + 1024 channels). Stage 4: train **seg4 + head**. Previous segments are frozen each stage. We report *Test (real)* each epoch and *Test (aux)* in Stages 1–3.

## B.5 GPU Memory Measurement Protocol

For all CNN memory measurements (ResNet-18 and ResNet-50), we instrument the training scripts using PyTorch's CUDA memory APIs to report peak allocated and reserved memory per epoch. Specifically, at the beginning of each epoch we call

```
torch.cuda.reset_peak_memory_stats(device)
```

and at the end of the epoch we record

```
peak_alloc    = torch.cuda.max_memory_allocated(device)
peak_reserved = torch.cuda.max_memory_reserved(device)
```

These quantities are reported in bytes by PyTorch; in all tables we convert them to GiB via division by $1024^3$.

**Allocated vs. reserved.** The *allocated* value measures the maximum amount of memory actually used by tensors during the epoch. We therefore treat peak allocated memory as our primary metric for model memory demand. The *reserved* value includes additional memory held by the CUDA caching allocator that may not be immediately released back to the driver; as a result, it can remain higher than strictly necessary or fluctuate across epochs due to allocator behavior. For clarity, we focus on peak allocated memory in the main text and tables, and we report reserved memory only for diagnostic purposes.

**Measurement setup.** All measurements are taken:

- on a single GPU, with `CUDA_VISIBLE_DEVICES` set to isolate the device,

- with batch size 256, mixed-precision (AMP) enabled, and AdamW optimization,

- using the same data pipeline (transforms, normalization, and CIFAR-10 splits) across E2E, LW, and SegProp.

For E2E runs, we report the steady-state peak allocated memory after the first warm-up epoch, which typically stabilizes for subsequent epochs. For SegProp stages, we report the per-stage peak allocated memory over a single epoch in steady state (once the CUDA allocator has warmed up). All reported values correspond to the maximum of `max_memory_allocated` over the epoch of interest.

### B.6 Auxiliary Networks (All Models & Regimes)

**Design goal.** Auxiliary modules ("aux") are minimal adapters that align intermediate features with the objective used in each training regime: (i) in **SegProp**, aux modules produce the *exact tensors* consumed by the global head (maintaining a single global objective at every stage); (ii) in **Layer-Wise (LW)**, aux modules produce a canonical feature map that is consumed by a *local* classifier (GAP + Linear), providing a stage-local training signal. To unify code and reduce parameters, we use *one-convolution adapters* with stage-specific strides that directly reach the canonical spatial size (7×7 for ResNets).

#### B.6.1 ResNet-18: Aux Designs

**SegProp (one-conv aux).** Let $x^{(s)}$ denote the output tensor of stage $s$ (seg1/seg2/seg3). We construct a *single* $1 \times 1$ convolution that: (1) downsamples spatially to $7 \times 7$ using stride $s\_7$, and (2) outputs 1024 channels that are split evenly into the required pair for the global head:

$$\underbrace{\text{Aux}^{(s)}(x^{(s)})}_{\in \mathbb{R}^{B \times 1024 \times 7 \times 7}} \xrightarrow{\text{split}} \mathbf{I}, \mathbf{P} \in \mathbb{R}^{B \times 512 \times 7 \times 7}.$$

The mapping is:

$$h = \sigma\big(\text{BN}(\text{Conv}_{1\times1}^{s\_7}(x^{(s)}))\big), \quad \mathbf{I} = h[:, 1{:}512, :, :], \quad \mathbf{P} = h[:, 513{:}1024, :, :],$$

where $\sigma$ is ReLU and $s\_7$ is the stride needed to reach $7 \times 7$ from the stage's spatial size:

$$s\_7 = \begin{cases} 8 & \text{for seg1 } (56{\to}7), \\ 4 & \text{for seg2 } (28{\to}7), \\ 2 & \text{for seg3 } (14{\to}7). \end{cases}$$

Stage 4 uses the *real* **seg4** (layer4 pre-head transform), no aux.

**LW (one-conv aux + GAP + Linear).** For stage $s \in \{1, 2, 3\}$, the aux produces a canonical feature $\tilde{z} \in \mathbb{R}^{B \times 512 \times 7 \times 7}$ via a single $1 \times 1$ conv (stride $s\_7$ as above), then applies a local classifier:

$$\tilde{z} = \sigma\big(\text{BN}(\text{Conv}_{1\times1}^{s\_7}(x^{(s)}))\big), \quad \hat{\mathbf{y}}^{(s)} = \mathbf{W}\,\text{GAP}(\tilde{z}) + \mathbf{b}.$$

Stage 4 aux: identity mapping to GAP (input already $[512, 7, 7]$). The final "fc" stage trains the real head only.

#### B.6.2 ResNet-50: Aux Designs

**SegProp (one-conv aux for three-conv head).** For ResNet-50 SegProp with a three-convolution head, the global head expects an *identity* tensor $\mathbf{I} \in \mathbb{R}^{B \times 2048 \times 7 \times 7}$ and an input tensor $\mathbf{F}_{\text{in}} \in \mathbb{R}^{B \times 1024 \times 7 \times 7}$. The aux adapters are as defined above:

$$h^{(s)} = \sigma\big(\text{BN}(\text{Conv}_{1\times1}^{s\_7}(x^{(s)}))\big) \in \mathbb{R}^{B \times 3072 \times 7 \times 7},$$

$$\mathbf{I}_s = h^{(s)}[:, 1{:}2048, :, :], \qquad \mathbf{F}_{\text{in}}^{(s)} = h^{(s)}[:, 2049{:}3072, :, :],$$

with $s\_7 \in \{8, 4, 2\}$ for seg1/2/3 as above. Stage 4 uses the real $f_4$ to produce $(\mathbf{I}, \mathbf{F}_{\text{in}})$.

**LW (one-conv aux + GAP + Linear).** For $s \in \{1, 2, 3\}$, a single $1 \times 1$ conv (stride $s\_7$) maps the stage features to the canonical $[2048, 7, 7]$, followed by GAP and a linear classifier:

$$\tilde{z} = \sigma\big(\text{BN}(\text{Conv}_{1\times1}^{s\_7}(x^{(s)}))\big), \quad \hat{\mathbf{y}}^{(s)} = \mathbf{W}\,\text{GAP}(\tilde{z}) + \mathbf{b}, \quad \tilde{z} \in \mathbb{R}^{B \times 2048 \times 7 \times 7}.$$

Stage 4 aux: GAP + Linear on $[2048, 7, 7]$.

**Implementation notes.**

- **Parameterization.** All aux adapters use one $1 \times 1$ convolution (stride set to reach canonical spatial size), followed by BN and ReLU. Weights are He-initialized; biases are zeroed where present.

- **Stability.** To avoid numerical drift when segments are frozen, frozen modules are kept in `eval()` mode so BatchNorm running statistics do not update. We apply AMP and use unscale+clip before optimizer steps to avoid infrequent FP16 spikes.

- **SegProp vs LW.** In SegProp, aux outputs *replace* the role of later segments during early stages by feeding the *global head* directly (ResNets: $(\mathbf{I}, \mathbf{P})$ or $(\mathbf{I}, \mathbf{F}_{\text{in}})$). In LW, aux outputs feed a *local* GAP+Linear classifier with its own loss; the global head is only trained in the final stage.

# C  Architectures and Training Regimes: LLM fine-tune

This appendix describes the model architectures and the common data and optimization settings used in our experiments. We consider two decoder-only Transformer language models:

- **Llama-3.1-8B-Instruct** (32 decoder layers),

- **Mistral-Nemo-Instruct-2407** (40 decoder layers).

Both models are fine-tuned under three regimes: End-to-End (E2E), Layer-Wise (LW), and Segmented Propagation (SegProp). The codebase is shared across all regimes, and differences arise only from which layers are trainable and how gradients are routed.

## C.1  Common Data and Optimization Settings

**Tasks and Benchmarks.** We focus on two standard benchmarks:

- **MMLU** (Massive Multitask Language Understanding),

- **WinoGrande**.

For both tasks, evaluation is performed using the `lm_eval` harness in few-shot mode:

- **MMLU**: 5-shot, micro batch size 16, batch size 384 (Mistral-Nemo-Instruct-2407),

- **WinoGrande**: 5-shot, micro batch size 8, batch size 384 (Llama-3.1-8B-Instruct).

All reported metrics are obtained using the full fine-tuned model in standard forward mode (no auxiliary modules are used at inference).

**Data Loading and Tokenization.** All fine-tuning and evaluation runs use a unified data pipeline. The key components are:

- **Tokenizer:** we use

  ```
  AutoTokenizer.from_pretrained(model_id, trust_remote_code=True).
  ```

- **Chat-style formatting:** for instruction-style data, we rely on `tokenizer.apply_chat_template` to construct the packed input sequence from message lists.

- **Padding and special tokens:**

- – we set `tokenizer.pad_token = tokenizer.eos_token` when the model does not define a pad token,
- – BOS and EOS markers are standardized using constants `BOS`, `EOS`.

- **Sequence statistics:** we compute sequence length statistics (max, min, mean, percentiles) using `compute_stats` and `compute_stats_verbose` to verify that sequence lengths remain within model limits.

**Evaluation Functions.** For MMLU and WinoGrande we use task-specific metric functions integrated into the HuggingFace `Trainer`:

- **MMLU:** we enable a dedicated pipeline via:
  - – `configure_mmlu_choice_token_sets(tokenizer)` to define the answer choice tokenization,
  - – `set_mmlu_eval_active(True)` to activate MMLU-specific processing,
  - – `compute_acc_mmlu` for computing multiple-choice accuracy on the letter options.

- **WinoGrande:** we use `compute_accuracy` over the model's predictions on the two candidate spans.

During training, these functions are passed as `compute_metrics` to the HuggingFace `Trainer`. Logits can be preprocessed by `_preprocess_logits_for_metrics` to reduce memory and speed up metric computation.

**Optimization and Distributed Training.** Unless otherwise stated, all fine-tuning runs share the following optimization settings:

- **Optimizer:** AdamW (via HuggingFace / DeepSpeed)
  - – learning rate: `1e-6`,
  - – betas: $(0.9, 0.999)$,
  - – $\epsilon$: $10^{-8}$,
  - – weight decay: configured as "auto" or a small positive value as per the DeepSpeed configuration.

- **Learning rate schedule:** linear decay with warmup:
  - – warmup ratio typically set to 3% of total training steps.

- **Gradient clipping:** max gradient norm 1.0.

- **Precision:** bfloat16 is used whenever GPU hardware supports it; otherwise fp16 is used. The helper `detect_gpu_config()` automatically selects between bf16 and fp16.

**DeepSpeed and FSDP Configuration.** We employ DeepSpeed ZeRO-3 and, where applicable, PyTorch FSDP for memory-efficient training of the 8B and 40-layer models:

- **Per-GPU micro-batch size:** `train_micro_batch_size_per_gpu = 16`.

- **Gradient accumulation steps:** `gradient_accumulation_steps = 6`.

- **Effective train batch size:** `train_batch_size = 384` (for a reference GPU count; the driver scripts recompute accumulation steps based on the actual number of devices).

- **Precision:** `bf16.enabled = true`.

- **ZeRO-3 optimization:**
  - – `stage = 3`,

- – overlapping communication and computation (`overlap_comm = true`, `reduce_scatter = true`),
  - – parameter and optimizer state offload to NVMe at `/opt/dlami/nvme`,
  - – `gather_16bit_weights_on_model_save = true`.

- **Gradient checkpointing:**

  - – partition activations,
  - – contiguous memory optimization,
  - – CPU checkpointing enabled for further memory savings.

The training script initializes distributed training as follows:

- the process group is created with `dist.init_process_group(backend="nccl")`,

- each rank sets its CUDA device via `torch.cuda.set_device(local_rank)`,

- the model is loaded with `device_map = "auto"` or an explicit mapping when quantization is used.

**Quantization.** To reduce memory and allow efficient fine-tuning:

- We optionally load the base model in 4-bit or 8-bit mode using `BitsAndBytesConfig`:

  - – 4-bit: `load_in_4bit = True`, with a configurable compute dtype (bf16, fp16, or fp32),
  - – 8-bit: `load_in_8bit = True`.

Quantization is applied consistently across E2E, LW, and SegProp regimes so that differences in performance are attributable to the training regime and layer selection, not to changes in low-level optimization.

# D  Architectures and Training Regimes: LLaMA-70B

This appendix describes the model architecture, backbone segmentation, training regimes, and memory-management techniques used in our LLaMA-70B SegProp experiments. The implementation targets $8\times$ A100/H100 40 GiB GPUs and is designed to fit the full 70B model within the available device memory through a combination of FSDP sharding, optimizer-state CPU offloading, chunked loss computation, and lazy layer materialization.

## D.1  Model Architecture

We use a standard decoder-only Transformer with the following configuration:

- **Vocabulary size:** $V = 128{,}256$.

- **Hidden size:** $d = 8{,}192$.

- **Intermediate size (MLP):** $d_{\text{ff}} = 28{,}672$.

- **Number of decoder layers:** $L = 80$.

- **Attention heads:** $H = 64$ query heads, $H_{\text{kv}} = 8$ key/value heads (Grouped Query Attention, GQA, with 8 groups of 8 query heads each).

- **Head dimension:** $d_h = d/H = 128$.

- **Positional encoding:** Rotary Position Embedding (RoPE) with $\theta = 500{,}000$.

- **Normalization:** RMSNorm with $\varepsilon = 10^{-5}$ applied before each sub-layer (pre-norm).

- **Activation:** SwiGLU in the MLP sub-layer.

- **Sequence length:** $T = 2{,}048$ tokens.

Each decoder layer $\ell$ computes:

$$\mathbf{h}_\ell = \mathbf{h}_{\ell-1} + \text{Attn}(\text{RMSNorm}(\mathbf{h}_{\ell-1})), \qquad \mathbf{h}_\ell \leftarrow \mathbf{h}_\ell + \text{MLP}(\text{RMSNorm}(\mathbf{h}_\ell)),$$

where Attn uses GQA with causal masking and RoPE applied to queries and keys. The final output is produced by a shared RMSNorm followed by a linear LM head:

$$\mathbf{o} = \mathbf{W}_{\text{lm}} \text{RMSNorm}(\mathbf{h}_L) \in \mathbb{R}^{B \times T \times V}.$$

**Parameter count.** The total parameter count is approximately 70B, distributed as follows:

- Embedding table (`embed_tokens`): $V \times d = 128{,}256 \times 8{,}192 \approx 1.05\text{B}$ parameters.

- Each decoder layer: $\approx 0.856\text{B}$ parameters (attention projections + MLP).

- LM head: $V \times d \approx 1.05\text{B}$ parameters.

- Total: $\approx 70.6\text{B}$ parameters; in bfloat16, $\approx 141.2\,\text{GiB}$.

### D.2 Backbone Segmentation

We partition the 80 decoder layers into four segments: three *body segments* and one *Global Head (GH)* segment. The GH always comprises the last $D_{\text{gh}} = 4$ decoder layers (layers 76–79) together with the final RMSNorm and the LM head. The remaining 76 body layers are divided as evenly as possible among the three body segments, with the remainder distributed to the earliest segments:

| Segment | Layers | Layer count | Approx. size (bf16) |
|---|---|---|---|
| seg0 | layers $[0, 26)$ | 26 | $\approx 22.3\,\text{GiB}$ |
| seg1 | layers $[26, 51)$ | 25 | $\approx 21.4\,\text{GiB}$ |
| seg2 | layers $[51, 76)$ | 25 | $\approx 21.4\,\text{GiB}$ |
| GH | layers $[76, 80)$ + RMSNorm + LM head | 4 | $\approx 5.5\,\text{GiB}$ |

The embedding table (`embed_tokens`) is logically attached to seg0 and trained jointly with it during Stage 1.

**Global Head definition.** The GH consists of:

1. Four full decoder layers (layers 76–79), each with GQA attention and SwiGLU MLP.

2. A final RMSNorm.

3. A linear LM head $\mathbf{W}_{\text{lm}} \in \mathbb{R}^{V \times d}$.

Given the hidden state $\mathbf{x} \in \mathbb{R}^{B \times T \times d}$ produced by the preceding body segment, the GH computes:

$$\mathbf{x}_{\text{gh}} = f_{\text{GH}}(\mathbf{x}) \in \mathbb{R}^{B \times T \times d}, \qquad \mathbf{o} = \mathbf{W}_{\text{lm}} \text{RMSNorm}(\mathbf{x}_{\text{gh}}) \in \mathbb{R}^{B \times T \times V}.$$

The GH is shared and updated at every training stage.

### D.3 SegProp Training Regime

**Stage-wise schedule.** With $N = 3$ body segments, `seg_epochs = 1`, and `seg_overlap = 0`, the total training spans `seg_stride` $\times (N - 1) +$ `seg_epochs` $= 1 \times 2 + 1 = 3$ epochs. Segments are introduced one per epoch:

| Epoch | Active | Training | Frozen | Forward path |
|-------|--------|----------|--------|--------------|
| 0 | seg0, GH | seg0 + embed, GH | — | embed $\rightarrow$ seg0 $\rightarrow$ GH |
| 1 | seg0, seg1, GH | seg1, GH | seg0 | embed $\rightarrow$ seg0 $\rightarrow$ seg1 $\rightarrow$ GH |
| 2 | seg0, seg1, seg2, GH | seg2, GH | seg0, seg1 | embed $\rightarrow$ seg0 $\rightarrow \cdots \rightarrow$ seg2 $\rightarrow$ GH |

At each epoch the GH is always in `train()` mode and its optimizer is updated every step. Frozen segments are set to `eval()` so that RMSNorm running statistics do not update. Activations are detached between segments ($\mathbf{x} \leftarrow \mathbf{x}$.`detach()`) so that gradients do not flow into frozen segments.

**Freezing and `eval()` mode.** Frozen modules are set to `eval()` so that RMSNorm running statistics do not update; only the current-stage segment and the GH are in `train()` mode. We use mixed precision (AMP, bfloat16) throughout; no gradient scaler is needed for bf16.

### D.4 FSDP Baseline

For comparison, we also implement a standard FSDP training baseline (`train_fsdp`):

- **Sharding strategy:** `FULL_SHARD` (ZeRO-3 equivalent): parameters, gradients, and optimizer states are all sharded across 8 GPUs.

- **CPU offload:** `CPUOffload(offload_params=True)` moves sharded parameters and optimizer states to CPU between forward/backward passes.

- **Activation checkpointing:** applied to every `LLaMADecoderLayer` via `apply_activation_checkpointing` with `CheckpointImpl.NO_REENTRANT`.

- **Mixed precision:** `MixedPrecision(param_dtype=bf16, reduce_dtype=fp32, buffer_dtype=bf16)`.

- **Backward prefetch:** `BACKWARD_PRE`.

- **Optimizer:** a single AdamW instance over all parameters; no per-unit splitting.

- **Initialization:** rank 0 materializes each decoder layer lazily (one at a time, $\approx 1.7\,\text{GiB}$ per layer) and broadcasts to all ranks via `sync_module_states=True`; non-rank-0 ranks use `param_init_fn` to allocate empty GPU tensors.

### D.5 GPU Memory Measurement Protocol

For all LLaMA-70B memory measurements, we instrument the training scripts using PyTorch's CUDA memory APIs to report peak allocated and reserved memory per step. At the beginning of each epoch we call

```
torch.cuda.reset_peak_memory_stats(device)
```

and at each instrumented code point we record

```
alloc    = torch.cuda.memory_allocated(device) / (1024**3)
reserved = torch.cuda.memory_reserved(device) / (1024**3)
peak     = torch.cuda.max_memory_allocated(device) / (1024**3)
```

followed by `torch.cuda.synchronize(device)` to ensure all pending CUDA operations have completed before reading the counters.

**Allocated vs. reserved.** The *allocated* value measures the maximum amount of memory actually used by live tensors at that point. We treat peak allocated memory as our primary metric for model memory demand. The *reserved* value includes additional memory held by the CUDA caching allocator that may not be immediately released back to the driver; it can remain higher than strictly necessary due to allocator behavior and fragmentation.

## E Llama-3.1-8B-Instruct Architecture

**Backbone Structure.** We use a HuggingFace-style implementation of Llama-3.1-8B-Instruct with a decoder-only architecture. The model consists of:

- A token embedding matrix `embed_tokens` mapping vocabulary indices to $H$-dimensional vectors.

- A stack of $N = 32$ decoder layers:

$$\texttt{self.layers} = [\text{LlamaDecoderLayer}_0, \dots, \text{LlamaDecoderLayer}_{31}].$$

- A final RMSNorm `norm`.

- An LM head `lm_head` projecting from the hidden dimension $H$ to the vocabulary size $V$.

Each `LlamaDecoderLayer` contains:

- **Self-attention:** `LlamaAttention` with rotary positional embeddings and grouped key/value heads.

- **MLP:** `LlamaMLP` with a gated activation.

- **Normalization:** two `LlamaRMSNorm` instances, one before attention and one before the MLP.

The forward computation follows:

$$x_0 = \texttt{embed\_tokens}(input\_ids), \tag{E.1}$$
$$x_{i+1} = \text{LlamaDecoderLayer}_i(x_i), \quad i = 0, \dots, 31, \tag{E.2}$$
$$h = \texttt{norm}(x_{32}), \tag{E.3}$$
$$\text{logits} = \texttt{lm\_head}(h). \tag{E.4}$$

**Segmentation and Shared Head.** For SegProp and LW regimes, we conceptually partition the 32 layers into:

- a **base prefix** of early layers (e.g., layers 0 to $p-1$, with $p = 18$ in our main runs),

- a set of **intermediate layers** $p, \dots, N-3$,

- a **global head segment** consisting of the last two decoder layers and the LM head:

$$\text{Head} = (\text{LlamaDecoderLayer}_{N-2}, \text{LlamaDecoderLayer}_{N-1}, \texttt{lm\_head}).$$

This segmentation enables:

- **E2E:** all layers are jointly trainable, no segmentation is enforced.

- **LW:** only one layer (or a small segment) is updated at a time, while earlier layers are frozen.

- **SegProp:** at each step, one intermediate layer is trained jointly with the always-active global head segment, while other layers may be frozen or used in a forward-only capacity.

The implementation uses environment variables read in `LlamaModel` to determine which layers are active:

- `LAYER_IDX_TO_TRAIN` selects the current layer index $i$ to update.

- `OMIT_LAYER_IDXS` and `DEAD_LAYER_IDXS` specify, respectively, temporarily omitted and permanently pruned layers.

- `SEGPROP_MODE` selects among:
    - `"e2e"` (all layers trainable),
    - `"efficient_pruning_alg"` (SegProp stages),
    - `"efficient_pruning_alg_base_model"` (initial reduced model).

At the optimizer level, we use `unfreeze_layers_by_indices()` to set `requires_grad = True` only for:

- the layer given by `LAYER_IDX_TO_TRAIN` (during SegProp/LW stages), and

- the last two layers plus `lm_head`, when we treat them as the shared global head.

All other layers keep `requires_grad = False`, making the training behaviour consistent across E2E, LW, and SegProp while allowing us to isolate the contribution of specific layers.

## F  ViT/ImageNet Experimental Setup

This section describes the full experimental configuration used for the ViT/ImageNet results reported in Section 4.3 and Table 2.

**Architecture.**  We use ViT-B/16 with $224 \times 224$ inputs and $C = 1000$ output classes (ImageNet-1k). The encoder consists of $L = 12$ transformer blocks. The global head (GH) is defined as the last $d_{\mathrm{GH}}$ blocks together with the encoder layer-norm and the classification head; the remaining $L - d_{\mathrm{GH}}$ blocks form the body. In the notation of Section 3.3, $\mathrm{LL}^{(r)}$ consists of the last $d_{\mathrm{GH}}$ transformer blocks, the encoder layer-norm, and the linear classifier.

The default configuration uses $d_{\mathrm{GH}} = 3$, leaving 9 body blocks that are divided evenly among $S = 2$ body segments:

$$
\begin{aligned}
\mathrm{Seg}\,0: &\quad \text{blocks } [0, 5), \quad 5 \text{ blocks}, \\
\mathrm{Seg}\,1: &\quad \text{blocks } [5, 9), \quad 4 \text{ blocks}, \\
\mathrm{GH}: &\quad \text{blocks } [9, 12) + \mathrm{LN} + \text{head}, \quad 3 \text{ blocks}.
\end{aligned}
$$

The patch-embedding projection (`conv_proj`), class token, positional embedding, and encoder dropout are assigned to Seg 0 and trained jointly with it. The ablations in Figure 4 vary $d_{\mathrm{GH}} \in \{3, 4, 5\}$ and the number of body segments $S \in \{1, \dots, 6\}$.

**Data pipeline.**  Data loading uses NVIDIA DALI with per-rank sharding. The training pipeline applies, in order: (i) random resized crop to $224 \times 224$ (scale $[0.08, 1.0]$), (ii) random horizontal flip, (iii) output as uint8 CHW tensors. A GPU-side post-processing module (`PostDALITrainAugment`) then applies RandAugment (num_ops$= 2$, magnitude$= 9$) on the integer pixel values, normalises to float with ImageNet statistics ($\mu = [0.485, 0.456, 0.406]$, $\sigma = [0.229, 0.224, 0.225]$), and applies RandomErasing ($p = 0.25$, scale $[0.02, 0.2]$, ratio $[0.3, 3.3]$). The validation pipeline resizes the shorter side to 256, applies a $224 \times 224$ centre crop, and normalises with the same statistics.

**Mixup and CutMix.**  Each training batch is augmented with Mixup ($\alpha = 0.8$) or CutMix ($\alpha = 1.0$), selected with equal probability ($p_{\mathrm{switch}} = 0.5$); the combined augmentation is applied with probability $p_{\mathrm{mix}} = 1.0$. Label smoothing of $\epsilon = 0.1$ is applied to the cross-entropy loss.

**Optimisation.** Each body segment and the GH are optimised independently with AdamW ($\beta_1 = 0.9$, $\beta_2 = 0.999$, weight decay $\lambda = 0.05$). Parameters with ndim = 1 (biases, layer-norm scales) and the class token and positional embedding are placed in a no-decay group ($\lambda = 0$). Gradient norms are clipped to 1.0 before each optimizer step. Mixed-precision training uses bfloat16 (AMP); no gradient scaler is needed for bf16.

**Learning-rate schedules.** Each body segment uses an independent cosine LR schedule of length $T_{\text{seg}} = 100$ epochs, starting fresh (warmup $\rightarrow$ peak $\rightarrow$ min) when the segment is introduced:

$$\eta_k(\tau) = \begin{cases} \eta_{\max} \dfrac{\tau}{T_{\text{warm}}} & \tau < T_{\text{warm}}, \\ \eta_{\min} + \frac{1}{2}(\eta_{\max} - \eta_{\min})\left(1 + \cos\left(\pi \dfrac{\tau - T_{\text{warm}}}{T_{\text{seg}} - T_{\text{warm}}}\right)\right) & \tau \geq T_{\text{warm}}, \end{cases} \tag{F.1}$$

where $\tau$ is the segment's *local* epoch (0 on introduction), $T_{\text{warm}} = 5$, $\eta_{\max} = 3 \times 10^{-3}$, and $\eta_{\min} = 10^{-6}$. The GH uses a single cosine schedule spanning the full training ($T_{\text{total}}$ epochs) with the same $\eta_{\max}$ and $\eta_{\min}$. The GH learning-rate factor is $\gamma_{\text{GH}} = 1.0$ (i.e. the same peak LR as the body segments).

**Staged introduction schedule.** With $S = 2$ body segments, $T_{\text{seg}} = 100$, and overlap $\delta = 0$, the stride is $T_{\text{stride}} = 100$ and the total training spans $T_{\text{total}} = T_{\text{stride}} \times (S - 1) + T_{\text{seg}} = 200$ epochs:

| Global epochs | Active segments | Training segments |
|---|---|---|
| 0–99 | Seg 0, GH | Seg 0, GH |
| 100–199 | Seg 0, Seg 1, GH | Seg 1, GH (Seg 0 frozen) |

When `freeze_finished_segs=True` (default), a segment whose local epoch reaches $T_{\text{seg}}$ switches to forward-only mode: its blocks still participate in the forward pass (providing activations to the next segment), but no loss is computed, no backward pass is run, and its optimizer state is freed. Activations are detached between segments (`x.detach()`) so that gradients do not flow into frozen segments.

**Distributed training.** Experiments run on 8 GPUs using PyTorch DDP with the NCCL backend. The per-GPU batch size is 128, giving a global batch of 1024. Gradients are synchronised across ranks via asynchronous `all_reduce` followed by division by the world size; gradient clipping is applied after synchronisation. The GH optimizer is stepped once per body segment (not once per batch), so the GH receives one gradient update per active training segment per step.

**Validation.** Two validation protocols are used at the end of every epoch:

1. *Vanilla*: a single forward pass through the full model (`forward_patches` $\rightarrow$ all 12 blocks $\rightarrow$ LN $\rightarrow$ head), reporting top-1 and top-5 accuracy.

2. *SegProp ensemble*: the training forward flow is mirrored exactly. For each segment $(s, e)$, the activations from the previous segment are passed through blocks $[s, e]$, then through the GH blocks, LN, and head to produce per-segment logits. The final prediction is the mean of all segment logits. This ensemble is used to track SegProp-specific convergence.

Gradient checkpointing (`use_grad_checkpoint`) is disabled by default; when enabled, `torch.utils.checkpoint` is applied to each transformer block during the training forward pass, reducing peak activation memory by approximately 40–50% at the cost of ~30% extra compute.

**Hardware and software.** All ViT/ImageNet runs use 8×A100 (or equivalent) GPUs. The implementation uses PyTorch with `torch.set_float32_matmul_precision("high")` and `cudnn.allow_tf32=True`. `cudnn.benchmark=True` is enabled after the backbone is moved to `channels_last` memory format.

# G   Mistral-Nemo-Instruct-2407 Architecture

**Backbone Structure.**   Mistral-Nemo-Instruct-2407 is a Mistral-style decoder-only Transformer. The architecture is analogous to Llama, with the following components:

- Token embedding layer `embed_tokens`.

- A stack of $N = 40$ decoder layers:

$$\texttt{self.layers} = [\text{MistralDecoderLayer}_0, \ldots, \text{MistralDecoderLayer}_{39}].$$

- Final RMSNorm `norm`.

- LM head `lm_head`.

Each `MistralDecoderLayer` includes:

- **Attention: `MistralAttention`**, a multi-head self-attention block with rotary positional embeddings and grouped key/value heads.

- **MLP: `MistralMLP`**.

- **Normalization: `MistralRMSNorm`** layers before attention and MLP.

The overall computation is:

$$x_0 = \texttt{embed\_tokens}(input\_ids), \tag{G.1}$$
$$x_{i+1} = \text{MistralDecoderLayer}_i(x_i), \quad i = 0, \ldots, 39, \tag{G.2}$$
$$h = \texttt{norm}(x_{40}), \tag{G.3}$$
$$\text{logits} = \texttt{lm\_head}(h). \tag{G.4}$$

**Segmentation and Shared Head.**   As with Llama, we segment the 40 decoder layers into:

- a **base prefix** (e.g., layers 0 to $p - 1$ with $p = 18$),

- **intermediate layers** $p, \ldots, N - 3$,

- a **global head segment**:

$$\text{Head} = (\text{MistralDecoderLayer}_{N-2}, \text{MistralDecoderLayer}_{N-1}, \texttt{lm\_head}).$$

The same set of environment variables (`LAYER_IDX_TO_TRAIN`, `OMIT_LAYER_IDXS`, `DEAD_LAYER_IDXS`, `SEGPROP_MODE`) control which layers are active and which regime (E2E, LW, SegProp) is applied.

For MMLU runs, the driver:

- initializes a configuration with:
  - early layers $0 \ldots p - 1$ and the last two layers active,
  - intermediate layers in the omit set,

- runs an initial reduced-model fine-tuning on MMLU,

- iteratively activates one additional intermediate layer at a time and fine-tunes it jointly with the shared global head segment,

- evaluates each stage on MMLU with `lm_eval` (5-shot).

At each stage, `unfreeze_layers_by_indices()` is used to unfreeze only the current layer of interest and the global head; all other layers remain frozen. This makes the comparison between regimes consistent and isolates the effect of the SegProp scheduling versus E2E fine-tuning.

