# OpenReview forum: "Unlocking The Power Of Layer-By-Layer Training And Fine- Tuning"
_TMLR — Accepted by TMLR_

### Review · Reviewer_i3VH · 2026-03-01

**Summary Of Contributions:**

This paper focus on tackling the information degradation of layer-wise (LW) training caused by the lack of persistent global supervision.
1. The authors proposed Segmented Propagation (SegProp), a two-stage training paradigm that is designed to introduce global supervision to the LW training. Specifically, in the first stage, a base prefix with last-layers are jointly trained to establish a strong task signal. In the second stage, intermediate blocks are iteratively trained one at a time, each jointly with the same global head, with previously trained blocks frozen. ,
2. By design, the proposed SegProp reduces peak activation memory and optimizer state relative to E2E training, as less trainable parameters are involved at each stage.
3. The authors evaluate the performance of SegProp on ResNet-18/50 and LLM fine-tuning (Mistral-Nemo, Llama3.1-8B), and show that SegProp substantially narrows the gap to E2E on ResNet-18/50 and on LLM fine-tuning.

**Additional Comments:**

NA

**Audience:**

Yes

**Audience Explanation:**

In the era of LLM, any progress made in reducing the memory barrier / IO barrier / computation consumption of excessively large models is important.

**Claims And Evidence:**

No

**Claims Explanation:**

The empirical results reported are accurate and clearly presented (accuracy numbers, memory tables, learning curves), and they support the claim that SegProp improves over standard LW and approaches E2E on the settings tested.

However, several central claims are not supported by convincing evidence.

1. Claim that “persistent global supervision” addresses the core failure of LW.  The paper motivates this using prior work on information degradation and HSIC (Sakamoto & Sato; Tishby et al.). The *design* of SegProp is consistent with this story (one shared task loss via the final layers). Yet the **theoretical link is not substantiated**: there is no formal connection to the information-bottleneck principle or to the information plane, and **no analysis of how the SegProp optimum relates to the E2E optimum**, e.g., whether the gap can be bounded or closed under what conditions.

2. Claim that “This holds across both CNN backbones and LLM fine-tuning, suggesting that a persistent head is a robust mechanism
for preserving task-relevant information in segmented training regime*. Within the paper’s experiments (ResNet-18/50, Mistral-Nemo, Llama3.1-8B), the results are consistent. But training-from-scratch is **only on CIFAR-10** with **ResNet-18 and ResNet-50**; there is no ImageNet, no ViT, no systematic variation in architecture or scale (the Limitations state they “do not study extremely deep transformers, mixture-of-experts or multimodal architectures, or large-scale pretraining from scratch”). So the claim of a **broadly applicable** or **robust** training paradigm is **not supported by sufficient evidence**—the paradigm is not validated on larger or more diverse setups (e.g. ViT, scale, other backbones).

**Requested Changes:**

1. Theoretical evidence and analysis on how the current optimization objective reflect the information bottle neck principle.
2. More pretraining experiments on larger scale datasets and larger models architectures. Otherwise, the findings of these papers are constrained on small scopes.
3. Run-time complexity and training FLOPs need to be reported to show the memory reduction comes at an acceptable compromise.
4. Regards fine-tuning experiments, the comparison with LoRA-variants is necessary. This helps provide guidance on when to choose SegPro and when to use LoRA in practice.

---

> ### Author Response · Authors · 2026-03-31
> **rebuttle**
>
> We thank the reviewer for the careful reading and detailed suggestions. We agree that several of our original statements required clearer scope, stronger large-scale evidence, and more explicit compute accounting. The revised manuscript addresses each requested item as follows.
>
> 1) “Persistent global supervision” as the core failure mode of LW; IB/HSIC theory not substantiated.
>
> We agree that our submission should not be interpreted as providing a formal information-bottleneck (IB) or HSIC-based theory. In the revision, we explicitly downgrade IB/HSIC to qualitative motivation, and we avoid implying a formal connection to the information plane or guarantees about the SegProp vs. E2E optimum. In Sec. 2.1 we now state that the information-bottleneck viewpoint is used only as an interpretive lens (not measured), and we rephrase the mechanism claim as a hypothesis tied to empirical observations (“improved intermediate-layer task performance across depth/stages”), while emphasizing that we do not provide a formal information-theoretic analysis. Specific example, “We hypothesize that persistent task-level supervision can reduce premature stage-local specialization; empirically, we observe improved intermediate-layer task performance across depth/stages. We emphasize that we do not provide a formal information-theoretic analysis of this effect; rather, we demonstrate an empirical training design that consistently reduces the LW–E2E gap across the evaluated settings.”  in section 2.1 .Empirically, the paper supports the operational claim: keeping a persistent head and a single task loss consistently narrows the LW–E2E gap in the evaluated settings, across staged CNN training, ViT/Imagenet full training and LLM fine-tuning diagnostics across depth.
>
> 2) “Robust across CNNs and LLMs” / lack of large-scale and diverse pretraining evidence.
>
> We agree that broad robustness claims require evidence beyond small-scale CNNs. The revision extends evaluation to ImageNet-scale vision training and larger-scale systems analysis:
> * ViT on ImageNet-1k: Sec. 4.3 and Appendix F add ImageNet-scale ViT training with controlled ablations over segmentation granularity and global-head (GH) depth, reporting top-1 accuracy and explicitly quantifying an accuracy–time–memory frontier.
> * ResNet-101 on ImageNet-1k (memory-focused study): Sec. 4.4 / Table 1 report peak memory for ResNet-101/ImageNet under E2E vs SegProp and show how checkpointing interacts with segmentation in an activation-dominated regime.
> * LLaMA-70B feasibility: Sec. 4.5 provides a system-level study on 8×40GiB GPUs with measured memory traces and a direct baseline comparison to FSDP+CPU offload, demonstrating the method’s practicality at much larger scale.
>
> To avoid over-generalization, the revision also keeps a clear Limitations statement: we do not claim coverage of extremely deep transformers, MoE/multimodal models, or large-scale pretraining from scratch beyond the evaluated settings.
>
> 3) Runtime complexity / training FLOPs and the memory–compute compromise
> We agree this must be explicit. The revision adds both wall-clock time and FLOPs accounting:
> * Wall-clock time at ImageNet scale: Table 2 reports Train Time (hours) along with peak GPU memory and final Acc@1 for ViT/ImageNet across segmentation/GH configurations, and Sec. 4.3 discusses the resulting trade-off (e.g., deeper GH can recover vanilla accuracy at increased time).
> * FLOPs analysis at 70B scale: Sec. 4.6 derives per-sequence FLOPs for FSDP vs SegProp and reports the total overhead per effective pass (SegProp total vs FSDP), making the compute–memory trade-off transparent
> * Measured time/throughput at 70B: Sec. 4.5 reports measured throughput and training time normalization (SegProp’s staged epochs correspond to one effective pass through the body), complementing the FLOPs analysis with real runtimes.
>
> 4) LoRA baseline for fine-tuning guidance.
>
> We agree that LoRA is an important practical baseline. In the revision, we (i) explicitly state that PEFT methods such as LoRA are highly effective, (ii) clarify SegProp is not positioned as a replacement because it targets a different constraint (depth-wise gradient sparsity / segmented backprop), and (iii) note SegProp can be composed with LoRA by applying LoRA within the active segment and/or persistent head. We also acknowledge as a limitation that we do not provide a systematic LoRA comparison in the current submission. Specifically we added “Parameter-efficient fine-tuning (PEFT) approaches such as LoRA are highly effective for LLM adaptation under constrained compute and memory budgets. SegProp is not positioned as a replacement for PEFT, since the two techniques address different axes of the training trade-off. In practice, SegProp can be composed with PEFT by applying LoRA within the trainable segment and/or within the persistent global head, if desired.” in section 4.7.

---

> > ### Comment · Reviewer_i3VH · 2026-05-06
> >
> > Thanks for your detailed response and revision. The rebuttal improves clarity of the scope of this paper.

---

### Review · Reviewer_651R · 2026-03-05

**Summary Of Contributions:**

**Summary**

The paper proposes Segmented Propagation (SegProp), a training method that aims to combine the memory efficiency of layer-wise (LW) training with the strong performance of end-to-end (E2E) training. LW training reduces memory usage but often degrades performance because layers are optimized using local objectives rather than the final task loss. SegProp addresses this limitation by maintaining a persistent global head during segmented training, allowing all segments to be optimized using a consistent task-level objective. Experiments on convolutional networks and LLMs show that SegProp significantly outperforms LW training and achieves performance close to E2E training while requiring substantially less peak GPU memory.

**Strengths**
1. The work is well motivated, clearly identifying the limitations of LW training and proposing the intuitive idea of a persistent global head to address them.
2. Empirical validation across multiple architectures shows that SegProp can outperform LW training and generally achieve performance on par with E2E training.
3. The paper provides quantitative evidence that SegProp requires significantly less peak memory compared to E2E training.
4. Overall, the paper is well written: it explains the key ideas clearly and is easy to follow.

**Weaknesses**
1. While the empirical results show that SegProp achieves better performance than LW training, the paper does not analyze the differences between the features learned under the two training regimes. For example, the last line of Subsection 2.1 states that “By providing consistent global supervision throughout training, SegProp encourages intermediate layers to learn general, transferable features rather than prematurely specializing for classification,” but no empirical validation is provided to support this claim.
2. Although memory savings are quantified, the paper does not report the total GPU-hours or wall-clock training time required for sequential segmented training. As a result, it remains unclear whether SegProp is computationally efficient overall.
3. The paper does not compare SegProp against strong alternatives such as LoRA, adapters, or partial fine-tuning, which are common baselines for memory-efficient LLM fine-tuning.

**Audience:**

Yes

**Audience Explanation:**

The findings of this work are likely to interest researchers and practitioners working on efficient training of deep neural networks. It addresses an important practical challenge, reducing the high memory requirements of end-to-end training, while maintaining strong model performance.

**Broader Impact Concerns:**

I do not have any ethical concerns regarding this work.

**Claims And Evidence:**

Yes

**Claims Explanation:**

The claims made in this work are generally supported by clear and convincing empirical evidence. The paper evaluates SegProp across multiple architectures, including convolutional networks and LLMs, demonstrating that it consistently outperforms standard LW training while achieving performance close to E2E training. In addition to accuracy improvements, the authors provide quantitative measurements of peak GPU memory usage, showing that SegProp significantly reduces memory requirements compared to E2E training. The experiments are conducted across different settings and model sizes, which strengthens the credibility and generality of the results. Overall, the combination of performance comparisons and memory analysis provides sufficient empirical support for the paper’s main claims.

**Requested Changes:**

1. The presentation could be improved in the following ways:
   * The second paragraph of the introduction uses the abbreviations LW and HSIC without first defining their full forms.
   * The paper should provide clearer interpretations of Figure 2(b–c) and Figure 3(c).
   * It is unclear what is meant by “global head size,” as mentioned in Figure 3(b).

---

> ### Author Response · Authors · 2026-03-31
> **rebuttle**
>
> Thank you for the constructive feedback. We address each weakness and presentation request in the revised manuscript as follows.
>
> (W1) No analysis of feature differences / “transferable features” claim.
> We agree the earlier wording could be read as a mechanistic claim. In the revision, we explicitly frame this as a hypothesis and tie it to what we empirically measure, stating: “We hypothesize that persistent task-level supervision can reduce premature stage-local specialization; empirically, we observe improved intermediate-layer task performance across depth/stages,” and we also emphasize that we do not provide a formal information-theoretic analysis.
> Concretely, intermediate-layer behavior is supported by depth/stage diagnostics: staged CNN trajectories (Sec. 4.1–4.2) and layer-index evaluation curves for LLM fine-tuning (Fig. 5–6) show SegProp’s improved task performance across depth relative to LW.
>
> (W2) Total GPU-hours / wall-clock cost of sequential segmented training.
> The revision reports wall-clock training time explicitly in the ImageNet-scale setting: Table 2 includes final Acc@1, Train Time (hours), and peak GPU memory across segmentation granularity and global-head (GH) depth, and Sec. 4.3 discusses the resulting accuracy–time–memory frontier.
> For LLaMA-70B analysis (not full training), we provide measured training times/throughput and a detailed FLOPs analysis quantifying SegProp’s compute overhead vs FSDP (including the total per effective pass).
>
> (W3) No comparison to LoRA/adapters/partial fine-tuning for memory-efficient LLM FT.
> We clarify scope and positioning: PEFT methods such as LoRA are highly effective; SegProp is not presented as a replacement since they optimize different axes (parameter-efficient adaptation vs depth-wise gradient sparsity). We explicitly note SegProp can be composed with LoRA by applying it within the trainable segment and/or persistent head. We also state as a limitation that we do not provide a systematic LoRA comparison.
>
> Requested changes:
> Define LW and HSIC on first use.
> The introduction defines layer-wise (LW) and expands Hilbert–Schmidt Independence Criterion (HSIC) before using the abbreviations.
>
> Clearer interpretation of Fig. 2(b–c) and Fig. 3(c).
> Sec. 4.1.3 now explains that Fig. 2(b–c) are stage-concatenated, piecewise curves; slope changes reflect switching the trainable segment (and in LW, the auxiliary head), and dots mark per-stage best checkpoints.
> Sec. 4.2 adds an explicit interpretation of Fig. 3(c) as incremental gains across SegProp stages under the same persistent head, contrasting with LW where supervision changes across stages.
>
> Clarify “global head size.”
> We define “global head size/depth” operationally: in ResNet-50 the “1 conv / 2 conv / 3 conv” variants specify how many of the final bottleneck conv blocks are included in the persistent global head; the exact decomposition is formalized in Appendix B

---

> > ### Comment · Reviewer_651R · 2026-04-01
> >
> > Thank you for addressing the concerns and incorporating my feedback.

---

### Review · Reviewer_KFBD · 2026-03-21

**Summary Of Contributions:**

The paper proposes an improvement to layer-wise training, SegProp. In traditinoal layer-wise training, one iteratively trains a layer at a time while freezing the others. Compared to the standard end-to-end training, layer-wise training saves memory during training but often results in a significant drop in accuracy. The authors argue this is because it suffers from information degradation as the layers are optimized against a local auxillary head, and that local supervision fails to preserve task-relevant information throughout the architecture. This work addresses this by including a persistent global head (final layers) that are active throughout the entirety of the segmented training. It uses light-weight training only adaptors to match the shapes between the current layer and the global head. The method shows that it recovers accuracy better than layer-wise training, tested on ResNet, Llama-3.1 and Minstral-Nemo.

**Audience:**

Yes

**Audience Explanation:**

Training efficiency continues to be a struggle for training large models, and it has become increasingly prohibitive for individuals and smaller labs without access to HPC. It would be of interest to resource-constrained researchers or individuals interested in fine-tuning LLM on a lower-end GPU setup. Even if the total training time (wall-clock) is longer due to the sequential stages, the ability to trade training time for a lower memory ceiling is a highly relevant trade-off that many practitioners are currently seeking.

**Claims And Evidence:**

Yes

**Claims Explanation:**

SegProp is empirically benchmarked against end-to-end and layer-wise training paradigms on multiple architectures and datasets.

- It shows a measurable improvement in accuracy against traditional layer-wise training by about 4-5% on Cifar10 and ResNet18/50.

- It scales to LLM, showing improved MMLU over baseline.

- Peak memory reduction is tracked, supporting the smaller memory footprint claim.

- The inclusion of HSIC analysis provides a theoretical grounding why this method works.

**Requested Changes:**

- The core concept is well demonstrated. However, the empirical evidence relies on smaller benchmarks. It would benefit from bigger models (ResNet101/152, Llama2-30B etc) and datasets (imagenet etc) where memory saving would be more transformative to average researchers and individuals. Imagenet's 1000 classes would confirm that the global head can effectively supervise.

- It could use more baselines. Having just the vanilla layer-wise may not be sufficient to demonstrate SOTA performance. Consider comparing to recent, up-to-date layer-wise training papers.

- To allow practitioners clearly weight the trade-off - how much extra training time is required to achieve the memory savings, consider comparing overall FLOP and required wall clock time.

---

> ### Author Response · Authors · 2026-03-31
> **rebuttle**
>
> We thank the reviewer for the thoughtful suggestions. We agree that demonstrating SegProp at larger scale, broadening comparisons, and quantifying time/compute trade-offs are important; the revision substantially expands all three aspects.
>
> We have extended the evaluation to ImageNet and to substantially larger model scales (including a 70B feasibility study), specifically to address this concern.
> * ViT on ImageNet: We report top-1 validation accuracy under SegProp and show how a persistent global head can supervise the 1000-way task, including controlled ablations over segmentation granularity and global-head depth (Fig. 4, Table 2; setup in App. F)
> * Much larger LLM regime: we include a LLaMA-70B feasibility study on 8×40GiB GPUs with measured memory traces and throughput, directly targeting the memory-limited regime.
>
> Regarding the more baselines request:
> * We add contextual comparison to AugLocal on ResNet-101/ImageNet in terms of memory-reduction magnitude (section 4.4), and we extend Related Work with recent staged/layer-wise work (section 2.4, PGT)
>
> To let practitioners weigh memory savings vs extra compute/time, we now report both wall-clock training time and FLOPs analyses in the revised manuscript.
> * Wall-clock + memory + accuracy frontier (ViT/ImageNet): Table 2 reports Final Acc@1, total training time (hours), and max GPU memory (GiB) across SegProp configurations, making the trade-off explicit.
> * FLOPs accounting (LLaMA-70B): We provide a detailed FLOPs breakdown for FSDP vs SegProp, showing SegProp’s total compute overhead per effective pass (Sec. 4.6).
> * Measured throughput/time (LLaMA-70B): We additionally report measured throughput and normalized training-time comparison versus the FSDP+offload baseline (Sec. 4.5).

---

### Review · Reviewer_4pbY · 2026-03-31

**Summary Of Contributions:**

I am sorry for my late review. I am not sure if my late review will be taken into account by the editor. I am ok with both.

I like the idea of layerwise training; I like the breakdown of detailed peak memory, and I also like the various models and datasets being evaluated.

What I don't like is that some important baselines are missed:
1. https://arxiv.org/abs/2403.17919, https://aclanthology.org/2025.findings-acl.996.pdf. Both of them provide strong baselines for layerwise fine-tuning.
2. https://arxiv.org/pdf/2404.02827; a strong baseline for pre-training.

And I also share the concern the true performance of llm pre-training, rather than just memory measurement.

**Audience:**

Yes

**Audience Explanation:**

Efficient training of modern models are important.

**Claims And Evidence:**

Yes

**Claims Explanation:**

The results of Vision models are clear.

**Requested Changes:**

Adding more related work and baselines about layer-wise training.

1. https://arxiv.org/abs/2403.17919,

2. https://aclanthology.org/2025.findings-acl.996.pdf.

3. https://arxiv.org/pdf/2404.02827; a strong baseline for pre-training.

---

> ### Author Response · Authors · 2026-04-02
> **rebuttal (new vesrion uploaded v3)**
>
> Thank you for the feedback and for highlighting important missing baselines for layer-wise / memory-efficient adaptation and pre-training. We agree these are highly relevant, and we have updated the manuscript accordingly.
>
> We added two new related-work subsections that explicitly discuss and cite the requested works:
> * Layer-selective fine-tuning baselines. We added Section 2.5 (“Layer-selective fine-tuning”) covering LISA and OWS, describing their layer-sampling/freezing strategies for memory-efficient adaptation and clarifying their relation to SegProp. We also note that these methods are complementary to SegProp: they focus on which layers to update, whereas SegProp focuses on how to stage optimization while preserving a single end-task loss path via a persistent global head, and we note potential composability (e.g., applying layer selection within the trainable set of SegProp)
>
> * We added Section 2.6 (“Memory-efficient full-parameter optimization”) covering BAdam, describing its block-coordinate descent (BCD) with Adam-style updates and clarifying how it differs from SegProp.
> In particular, we explain that BAdam primarily changes the optimizer/update strategy via block-wise BCD-style updates, while SegProp changes the training schedule and supervision path by keeping a persistent global head on the loss path and training one segment jointly with this head at each stage. We also note potential composability (e.g., using a block-wise optimizer within the parameters trained by SegProp)
>
> We agree with the concern that pre-training quality (e.g., perplexity) is distinct from memory feasibility. To avoid overstating evidence, we clarified in the manuscript Limitations that our LLaMA-70B section is a system-level feasibility and compute–memory trade-off study (max feasible batch size, measured memory traces, FLOPs accounting) and does not claim improved pre-training quality relative to end-to-end training.

---

### Author Response · Authors · 2026-04-02
**added new revision post new review**

Thanks again to all reviewers for your feedback and suggestions.
I recently received a new review and submitted a new revision.
For clarity here are the changes made in both 1st and 2nd revision:


Summary of changes (v2 → v3)

1) Related Work (new baselines added)
- Added Sec. 2.5 “Layer-selective fine-tuning”: discusses LISA and OWS, clarifies they are complementary to SegProp, and notes potential composability (layer selection within the active segment/head).
- Added Sec. 2.6 “Memory-efficient full-parameter optimization”: discusses BAdam, clarifies how it differs from SegProp, and notes potential composability (block-wise optimizer within the trainable segment/head).

2) Clarification of the 70B section’s scope (avoid over-claiming)
- Updated the Limitations/discussion text to state that the LLaMA-70B results are a feasibility + compute–memory trade-off study (batch size, memory traces, FLOPs) and do not claim improved pre-training quality (e.g., perplexity).

3) References
- Updated the reference list to include the newly cited works (LISA, OWS, BAdam).


Summary of changes (Original → v2)

1) Reframed the abstract and positioning
- The abstract was rewritten to center SegProp as “persistent global head + segmented training” and to broaden the scope beyond CIFAR/LLM fine-tuning (original emphasized HSIC framing and “modified LW enables E2E”; v2 emphasizes staged training with a persistent head and a broader accuracy–time–memory trade-off story).
2) Expanded the experimental scope substantially
- Added ImageNet-scale ViT experiments with explicit accuracy–time–memory trade-offs vs segmentation granularity and global-head depth (new Sec. 4.3, Fig. 4, Table 2 in v2).
- Added ImageNet CNN memory analysis on ResNet-101, including comparisons with/without gradient checkpointing and discussion of when segmentation adds savings beyond checkpointing (new Sec. 4.4 and Table 1 in v2).
- Added a system-level feasibility + memory trace study for LLaMA-70B on 8×40GiB GPUs, comparing FSDP+offload vs SegProp and reporting max feasible batch sizes and measured throughput (new Sec. 4.5 and Tables 3–4 in v2).
- Added a detailed FLOPs accounting comparing FSDP vs SegProp for LLaMA-70B, quantifying the compute–memory trade-off (new Sec. 4.6 and Table 5 in v2).

3) Related Work expanded and reorganized
- Added a Related Work subsection on Progressive Growth Transformers (PGT) and an explicit comparison to SegProp (new Sec. 2.4 in v2).
- Revised the “information bottleneck / HSIC” discussion to be more cautious and explicitly framed as qualitative motivation rather than a formal/measured analysis.
4) Method description and appendices updated for the broadened scope
- Updated the narrative in the problem setting/method sections to align with the broader evaluation scope (CNNs + ViT/ImageNet + LLM system feasibility) and to clarify the persistent global head and staged segment updates as the core mechanism.
- Added/expanded appendices to cover the new ViT/ImageNet setup and the LLaMA-70B system setup and measurement protocol (Appendix F for ViT/ImageNet and Appendix D for LLaMA-70B in v2).
5) Shifted the memory story from a small CIFAR-10 table to ImageNet + 70B analyses
- The original included a dedicated “ResNet-50/CIFAR-10 peak memory” table and discussion; v2 broadens the memory evidence with ImageNet-scale ResNet-101 and ViT trade-offs, plus 70B feasibility and FLOPs analysis.

---

### Decision · Action_Editor_Siom · 2026-05-30

**Recommendation:** Accept as is

**Audience:**

Yes

**Audience Explanation:**

Efficient training and fine-tuning of large neural networks are topics of broad interest to the machine learning and AI community.

**Claims And Evidence:**

Yes

**Claims Explanation:**

The paper presents a well-motivated solution to a known limitation of layer-wise optimization. The empirical evaluation is comprehensive, covering both vision models and large language models, and consistently demonstrates the effectiveness of the proposed method.